# Thermal migration towards constructing W-W dual-sites for boosted alkaline hydrogen evolution reaction

Zhigang Chen[1,5], Yafeng Xu[2,5], Ding Ding[1], Ge Song[1], Xingxing Gan[1], Hao Li[1], Wei Wei[1], Jian Chen[1], Zhiyun Li[1], Zhongmiao Gong[1], Xiaoming Dong[1], Chengfeng Zhu[1], Nana Yang[1], Jingyuan Ma[3], Rui Gao[4], Dan Luo[4], Shan Cong[1], Lu Wang [2], Zhigang Zhao [1✉] & Yi Cui [1✉]

Tungsten carbides, featured by their Pt-like electronic structure, have long been advocated as potential replacements for the benchmark Pt-group catalysts in hydrogen evolution reaction. However, tungsten-carbide catalysts usually exhibit poor alkaline HER performance because of the sluggish hydrogen desorption behavior and possible corrosion problem of tungsten atoms by the produced hydroxyl intermediates. Herein, we report the synthesis of tungsten atomic clusters anchored on P-doped carbon materials via a thermal-migration strategy using tungsten single atoms as the parent material, which is evidenced to have the most favorable Pt-like electronic structure by in-situ variable-temperature near ambient pressure X-ray photoelectron spectroscopy measurements. Accordingly, tungsten atomic clusters show markedly enhanced alkaline HER activity with an ultralow overpotential of 53 mV at 10 mA/cm$^2$ and a Tafel slope as low as 38 mV/dec. These findings may provide a feasible route towards the rational design of atomic-cluster catalysts with high alkaline hydrogen evolution activity.

[1] Vacuum Interconnected Nanotech Workstation, Suzhou Institute of Nano-Tech and Nano-Bionics, Chinese Academy of Sciences, Suzhou 215123, China. [2] Institute of Functional Nano & Soft Materials (FUNSOM), Jiangsu Key Laboratory for Carbon-Based Functional Materials & Devices, Soochow University, Suzhou, Jiangsu 215123, China. [3] Shanghai Synchrotron Radiation Facility (SSRF), Shanghai Advanced Research Institute, Chinese Academy of Sciences, Suzhou 201204, China. [4] Department of Chemical Engineering, Waterloo Institute for Nanotechnology, Waterloo Institute for Sustainable Energy, University of Waterloo, Waterloo, ON N2L 3G1, Canada. [5] These authors contributed equally: Zhigang Chen, Yafeng Xu. ✉email: zgzhao2011@sinano.ac.cn; ycui2015@sinano.ac.cn

The electrocatalytic hydrogen evolution reaction (HER) in alkaline media is one of the most economical and effective routes for producing high-purity hydrogen gas that meets sustainability imperatives[1]. Tungsten carbide materials, featured by their Pt-like electronic structure, have long been advocated as potential replacements for the benchmark Pt-group metals in alkaline HER electrocatalysis[2]. Tuning the local coordination structure in tungsten carbides is meaningful and worthwhile because their HER behaviors are closely correlated with the coordinatively unsaturated structure of the W centers[3]. For example, tungsten single atoms coordinated via unsaturated W-C bonds can exhibit markedly improved activity with an ultralow overpotential at 10 mA/cm$^2$ ($\eta_{10} = 85$ mV) compared to that of stoichiometric WC nanoparticles ($\eta_{10} = 146$ mV)[3]. Usually, unsaturated coordination occurs in tungsten carbide nanoparticles (diameters < 5 nm) or tungsten single atoms because of the defective nature of their ultrasmall size[4,5]. To date, ultrasmall Mo-doped tungsten carbide nanoparticles (Mo-WC, ca. 5 nm) and carbon/nitrogen coordinated single tungsten atoms ($W_1N_1C_3$, ca. 1.2 Å) with theoretically superior alkaline HER activity have been synthesized via the pyrolysis of organic tungsten precursors[3,6], but their practical alkaline HER performances (i.e., $\eta_{10}$, Mo-WC: 179 mV; $W_1N_1C_3$: 85 mV) are far from those of noble metal catalysts, such as Pt ($\eta_{10} \approx 30$ mV)[7]. This can be understood in two ways that are neglected by theoretical simulations: (i) compared to Pt catalysts, the stronger OH* binding energies on tungsten carbides result in weak adsorption energies for the neighboring H* intermediates in alkaline HER electrocatalysis because of the strong trapping of free electrons by the vacant 5d orbitals of W atoms that are bonded to the OH* intermediates[8,9]; (ii) the strong affinity of OH* intermediates may induce the gradual oxidation of W sites into inert $W_xO_y$ species, which seriously impedes their catalytic activity and long-term durability[10]. Therefore, the exploration of ultrasmall tungsten-carbide catalysts with the ability to alter the energy barriers for the prior water dissociation step (i.e., the inherently sluggish step in alkaline media) and subsequent desorption of H*/OH* (* is the active site) intermediates is promising for addressing the challenging OH-induced oxidation problem and achieving excellent HER performances in alkaline electrolytes.

Recently, another unusual form of tungsten carbide: carbon coordinated tungsten atomic clusters (denoted W-ACs), formed between tungsten single atoms (denoted W-SAs) and tungsten carbide nanoparticles (denoted WC NPs) have drawn our attention. These materials have the potential to overcome the bottleneck of current tungsten-carbide-based alkaline HER catalysts, and show better electrocatalytic performance compared to previously reported tungsten single atoms and bulk tungsten carbide catalysts. The better performance of these clusters originates from the relationship between the local coordination structures and catalytic properties: (i) the presence of W-W bonds provides dual atomic sites for the adsorption of split H* and OH* intermediates, unlike the isolated sites in W-SAs, which is beneficial for the cleavage of H-OH bonds in the prior water dissociation step[11]; (ii) the coordinatively unsaturated W centers in the W-ACs can better regulate the adsorption state of H* and OH* intermediates compared to bulk WC NPs, thus accelerating their desorption kinetics[3,12]. Unfortunately, the formation temperature of carbide bonds is usually up to 700 °C[4]. Under such high-temperature condition, the subnanometer tungsten clusters are thermodynamically unfavorable and will aggregate into larger nanoparticles. Recently, Xie, et al. reported a pioneering work that demonstrated the synthesis of $Fe_2N_6$ clusters using single Fe atoms ($FeN_4$) as the parent materials via a thermal migration strategy, specifically, the desired $Fe_2N_6$ clusters were formed by treatment at 650 °C for 2 h under an Ar atmosphere[13]. Clearly,

such a single-atom thermal migration process can provide a feasible strategy for synthesizing ultrasmall tungsten clusters, as evidenced by the low-temperature (ca. 500 °C) thermal migration of tungsten single atoms observed on graphene substrates[14]. However, reports of the direct tracking of the transformation of single atoms to atomic clusters are still lacking, and this information is required to reduce the uncontrollability of the preparation method. Therefore, the development of an in-situ characterization technique to monitor the formation of W-ACs with Pt-like electronic structure is crucial.

Furthermore, despite the extensive investigation of tungsten-carbide materials for HER electrocatalysis, their Pt-like HER behaviors have only been documented in theoretical calculations using the Gibbs free energy of hydrogen adsorption ($\Delta G_H$) as the activity descriptor[15]. In these studies, both tungsten and carbon atoms in the tungsten carbides, and even the carbon/nitrogen atoms of the supports have been suggested to be active sites[3,10,16,17]. As can be seen, no consensus has been reached on the active sites for current tungsten-carbide catalyzed HER processes, and the discrepancy may be ascribed to the fact that the complicated local coordination structures in tungsten carbides yield several possible reaction pathways for the HER electrocatalysis, while no reliable experiment evidences support the calculations in practice. Therefore, the discrimination of the Pt-like HER behavior of coordination-controlled W-ACs using in situ detection techniques is both intriguing and valuable.

Here, we report the synthesis of tungsten atomic clusters anchored on P-doped carbon materials via a thermal-migration strategy using tungsten single atoms as the parent material, which is evidenced to have the most favorable Pt-like electronic structure by in-situ variable-temperature near ambient pressure X-ray photoelectron spectroscopy (NAP-XPS) measurements. Accordingly, the synthesized W-ACs exhibit markedly superior alkaline HER performance with an ultra-low overpotential of 53 mV at 10 mA/cm$^2$ and a small Tafel slope (38 mV/dec), as well as a high turnover frequency (TOF) value of 0.12 H$_2$ s$^{-1}$ at an overpotential of 50 mV. In contrast, inferior alkaline HER activity is observed in the control samples (W-SAs and WC NPs). The subsequent quasi in-situ HER observations and density functional theory (DFT) calculations show that the W-W dual-atoms of the W-ACs are the active sites for the adsorption of the produced H* and OH* intermediates, which exhibit a much less negative value of H* adsorption energy and easier OH* desorption behavior than those of W-SAs.

## Results

**Morphological characterization of tungsten atomic clusters.** The W-SAs were synthesized according to the previously reported protocols[18]. Briefly, in a typical synthesis, polydopamine (PDA, 3 g), polyethylene oxide-co-polypropylene oxide-co-polyethylene oxide (P$_{123}$, 3 g), 2-amino-2-hydroxymethyl-propane-1, 3-dio (Tris, 0.5 g), Na$_2$H$_2$PO$_4$·2H$_2$O (0.6 g) and 0.04 g Na$_2$WO$_4$·2H$_2$O (0.04 g) were dissolved in distilled water (30 mg) with continuous stirring over 6 h. The flower-like W precursors were then collected by centrifugation, washed with a mixture of distilled water and ethanol, and then dried at 60 °C (Supplementary Figs. 1, 2). Finally, W-SAs anchored on the P-doped carbon material were obtained by pyrolyzing the W precursors at 700 °C for 2 h under an Ar atmosphere, followed by the alkaline leaching in a 6 M KOH solution for 3 days. The W-ACs and WC NPs were obtained using W-SAs as the parent material via thermal migration strategy at 550 and 650 °C, respectively, under a mixture of Ar/H$_2$ (1 bar) atmosphere (Fig. 1a, for more details, see the methods section). Importantly, the P atom dopant in the carbon matrix contributes to regulate the electronic structures of W atoms in W-SAs and W-ACs because of its relatively weak

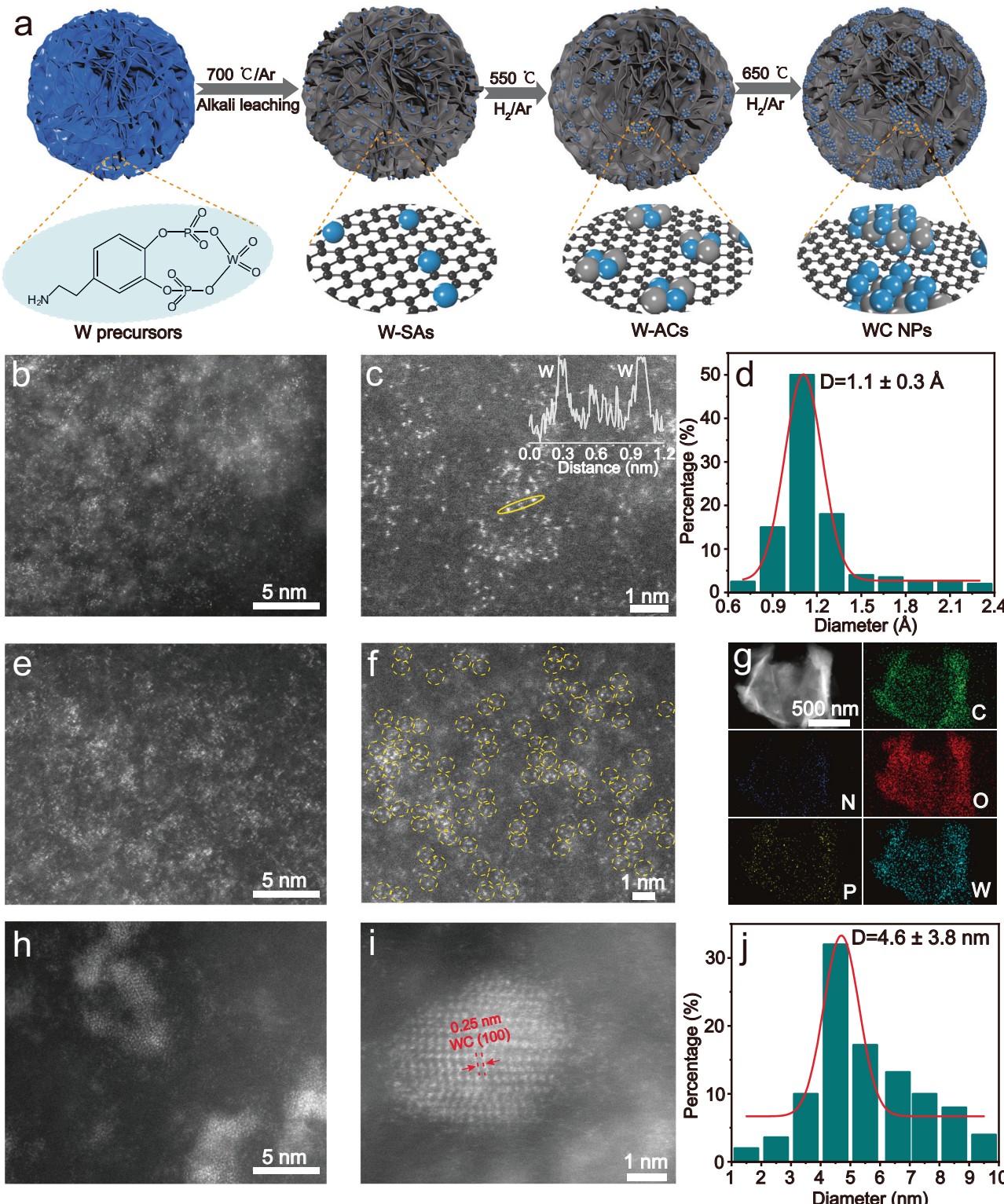

**Fig. 1 Morphological characterization of the synthesized W-ACs. a** Schematic illustration of the synthesis of W-SAs, W-ACs and WC NPs at H₂/Ar atmosphere (1 bar), where W atoms, carbide carbon, and graphitic carbon is represented by blue, gray, and black spheres, respectively. Typical HAADF-STEM images of (**b**) W-SAs, **e** W-ACs, **h** WC NPs. **c** Enlarged STEM image of W-SAs, inset shows the intensity profile along the tungsten atoms indicated by the yellow solid circle. **d** Statistical analysis of the lateral dimension of atomically dispersed tungsten atoms in (**b**) and Supplementary Fig. 4c. **f** Enlarged STEM image of W-ACs, with the W-W structures indicated by the yellow dashed circles. **g** STEM-EDS mapping of W-ACs showing the homogeneous distribution of C (green), N (deep blue), O (red), P (yellow) and W (cyan) elements. **i** High-magnification HAADF-STEM image of a WC NP showing the (100) facet with lattice fringe spacing of 0.25 nm. **j** Statistical analysis of the lateral dimension of WC NPs in (**h**) and Supplementary Figs. 5b, 7c.

electronegativity compared to that of carbon atom[19], as discussed later.

X-ray powder diffraction (XRD) pattern and Raman spectra were used for preliminary evaluation of the samples to determine whether an atomic dispersion of tungsten atoms anchored on the P-doped carbon materials (W-SAs and W-ACs) was obtained[20]. As shown in Supplementary Fig. 3a, the XRD patterns of W-SAs and W-ACs only contain peaks corresponding to the underlying carbon substrates (i.e., 21.3 and 43.5°) with no signals indicating tungsten-containing crystal phases. In contrast, an additional peak at 31.5° is obtained in WC NPs, which can be attributed to the (001) facet of the hexagonal tungsten carbide (JCPDS No. 51-0939)[10]. In the Raman spectra of W-SAs and W-ACs, only peaks at around 1341 and 1585 $cm^{-1}$ are observed, which can be attributed to the characteristic D and G bands of the underlying carbon layer, respectively, whereas WC NPs show a broad band corresponding to the W-C stretching mode at around 800 $cm^{-1}$ (Supplementary Fig. 3b)[10], implying that the atomically dispersed tungsten species aggregated into crystalline WC NPs after treatment at higher temperatures. High-resolution transmission electron microscopy (HRTEM) and atomic-resolution high-angle annular dark-feld scanning transmission electron microscopy (HAADF-STEM) were next used to observe the morphology of the as-obtained W-ACs and the control samples (W-SAs and WC NPs). Only carbon layers without any nanoparticles can be seen in the HRTEM image of W-SAs, because it is challenging to observe ultrasmall nanocrystals (i.e., diameters less than 1 nm) using TEM (Supplementary Fig. 4a, b). This result implies that the W atoms may be anchored in the carbon matrix in an atomically dispersed state. HAADF-STEM characterization was performed to investigate the atomic morphology of the as-obtained tungsten species. As shown in Fig. 1b, Supplementary Figs. 4c, 7a, numerous bright spots well-dispersed over the carbon substrates can be seen, and the intensity profile along the tungsten atoms indicated by the yellow solid circle (inset in Fig. 1c) confirms that the isolated bright spot is of single-atom size (ca. 1.1 Å), which is further verified by the lateral dimensions of 200 bright spots measured in Fig. 1b and Supplementary Fig. 4c, and the statistical analysis was presented in Fig. 1d, indicating high-quality atomically dispersed tungsten species have been obtained in W-SAs. After thermal treatment at 550 °C in an Ar/$H_2$ atmosphere, the W-SAs aggregated into W-ACs with sizes in the range of 0.2–1.2 nm (Fig. 1e, Supplementary Figs. 5a, 7b). The corresponding magnifed HAADF-STEM images clearly evidence that a large proportion of the bright spots are adjacent to each other and present a W-W dimer structure, as indicated by the yellow dashed circles (Fig. 1f, Supplementary Fig. 6). Meanwhile, the STEM-EDS mapping images show the homogeneous distribution of C, N, O, P, and W elements throughout the entire carbon matrix (Fig. 1g). In addition, according to the coupled plasma optical emission spectrometry (ICP-OES) analysis, the weight ratio of W atoms in W-ACs was determined to be of 10.49 wt%. After further treatment of the W-SAs up to 650 °C, a high density of nanoparticles with an average size of 4.6 nm were visible in the low-magnification HAADF-STEM image (Fig. 1h, j and Supplementary Figs. 5b, 7c), which was observed to have a lattice fringe spacing of 0.25 nm associated with the (100) facet of the crystallized WC phase (Fig. 1i). Thus, combining the information extracted from XRD pattern, Raman spectra, and HRTEM/HAADF-STEM characterization, we could conclude that the subnanometer W-ACs were indeed formed as an intermediate product during the thermally induced transition from W-SAs to WC NPs.

**Monitoring the formation of tungsten atomic clusters using NAP-XPS**. The formation of the intermediate W-ACs having Pt-like electronic structure can be dynamically monitored by probing the evolution of the electronic structures of W atoms from W-SAs

to WC NPs using in-situ variable-temperature NAP-XPS measurements with $H_2$ atmosphere at a pressure of 0.1 mbar (Fig. 2a and Supplementary Fig. 8). To detect the exact signal caused by the thermal migration of W atoms, W-SA parent materials were first treated by Ar etching under vacuum conditions to remove any possible remnants of the surface $WO_x$ passivation layer and graphitic carbon without impacting the W-C carbide structures[21,22], in which the Ar-etching treatment was proceeded in the analysis chamber of NAP-XPS system under Ar atmosphere ($5 \times 10^{-6}$ mbar), and the corresponding W 4f XPS signals were recorded to determine the optimized Ar-etching time under vacuum condition ($5 \times 10^{-9}$ mbar) (Supplementary Fig. 9). Subsequently, the treated sample was transferred to the analysis chamber for in situ variable-temperature NAP-XPS characterization, and the W *4f* XPS core-level signals were recorded at temperatures ranging from 400 to 600 °C (Fig. 2b). As can be seen, the initial state of W-SAs exhibits three pairs of peaks after deconvolution, in which the major peaks with lower binding energies at 31.5 and 33.6 eV can be attributed to the intrinsic W-C bonds, while the second pair of peaks at higher binding energies (35.8 eV for W $4f_{7/2}$ and 37.9 eV for W $4f_{5/2}$) are associated with the W-O bonds (Supplementary Table 1). In addition to the aforementioned two predominant pairs of peaks, another doublet with intermediate binding energies (32.6 and 34.8 eV) are also visible, which can be ascribed to the carbon/oxygen-coordinated tungsten species due to the trace amount of oxygen in W-C coordination structures (W-C(O))[23,24]. An obvious negative shift of the W $4f_{7/2}$ and W $4f_{5/2}$ peaks in W-O species towards lower binding energies was found at 400 °C (Supplementary Table 1), indicating the bonded oxygen atoms were removed by hydrogen gas as the temperature increased. The negligible changes in the intrinsic W-C signals were observed for the samples treated at 400 and 450 °C, implying that the W atoms were probably stable in a single-atom state on the carbon substrates[14]. When the temperature reached 500 °C, the intensities of W-C(O) and W-O signals decreased significantly (Supplementary Table 2), implying that they were transformed into carbon-coordinated tungsten species with lower valence state, which was also evidenced by the abruptly enhanced intensities of the low-valence W-C signals at 500 °C (Supplementary Table 2). Encouragingly, the intermediate W-C(O) species were almost absent, and the highest intensities of W-C signals with sharp peaks were detected in the sample treated at 550 °C (Supplementary Tables 2 and 3), indicating the W-ACs might have formed because of the thermal migration of tungsten atoms at high temperature[13,14]. However, the W-C signals became weaker when treated at temperatures higher than 550 °C (also confirmed by the controlled experiments in Supplementary Figs. 10, 11), indicating the surface of the generated W-ACs might have been simultaneously coated by a graphitic carbon layer[21], which would be transformed into the carbide carbon in the crystallized tungsten carbide phase at high temperatures, as evidenced by the C *1s* XPS spectra of tungsten species collected at 600 °C, which shows an obviously broad carbide C *1s* signal located at approximately 282.7 eV (Supplementary Fig. 12)[25]. Thus, the in situ NAP-XPS results disclose that the formation of desired W-ACs is probably confined to an extremely narrow temperature range of 550–580 °C. Finally, the corresponding valence bands of the tungsten species were also recorded to evaluate the Pt-like electronic structure obtained in the synthesized W-ACs. To facilitate comparison, the valence bands of single-crystalline Pt (111) and W foil were measured in Fig. 2c, and the valence bands of bare Ni foam and pristine P-doped carbon materials coated with Ni foam (C@Ni foam) were also collected to eliminate any interference of the substrates (Supplementary Fig. 13). All tungsten species exhibit typically metallic feature with valence band profiles crossing the Fermi level, where

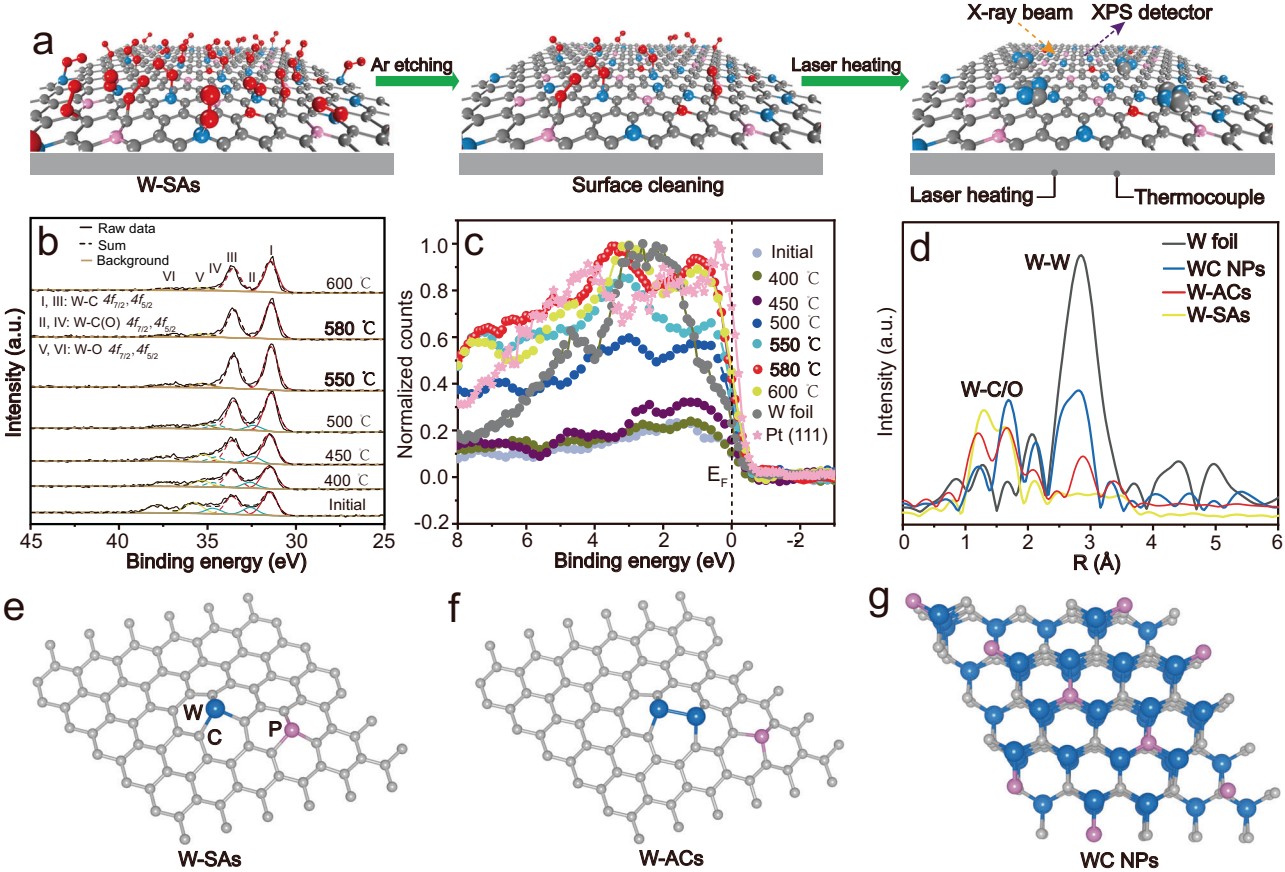

**Fig. 2 Structural characterization of the synthesized W-ACs. a** Schematic illustration of the in-situ variable-temperature NAP-XPS measurements at $H_2$ atmosphere (0.1 mbar), where O, C, P and W atoms are represented by red, gray, pink, and blue spheres, respectively. **b** W $4f$ core-level XPS spectra for tungsten species treated at temperature ranging from 400 to 600 °C. **c** XPS valence band spectra of various tungsten species and the reference samples of W foil and Pt (111). **d** FT-EXAFS spectra at the W $L_3$-edge of W-SAs, W-ACs, WC NPs, and W foil. Typical atomic models of (**e**) W-SAs, **f** W-ACs. **g** WC NPs. W (blue), C (gray), and P (pink).

the density of states near the Fermi level is very sensitive to the local electronic structure of W atoms[26]. The density of states near the Fermi level (0–2 eV) for all tungsten species treated above 500 °C are more intense than that of W foil reference, indicating the formation of W-W bonds endows the coordinatively unsaturated W centers with a higher d-band electronic density of states, thus resulting in a higher carrier density, which benefits the charge transfer during electrocatalytic reaction[4]. This is consistent with the result of a previous report of Pt atomic clusters[27]. Specifically, the tungsten species treated at 580 °C have the most favorable Pt-like electronic structure regarding both the density of states and peak features near the Fermi level. Thus, these results suggest that tungsten carbide materials with electronic structures similar to that of Pt were indeed achieved in W-ACs.

**Characterization of the coordination environment in the tungsten atomic clusters**. The W-SAs, W-ACs, and WC NPs were further examined by X-ray absorption near-edge structure (XANES) and Fourier-transformed extended X-ray absorption fine structure (FT-EXAFS) analysis. The white-line intensities for all tungsten species are higher than that of the W foil reference (Supplementary Fig. 14), suggesting that all tungsten species are in oxidation states[3,9]. Meanwhile, the shifting of the adsorption edge position can be used to identify subtle changes in the valence states of tungsten atoms in the as-obtained W-SAs, W-ACs and WC NPs[28]. The adsorption edge position exhibit a negative shift

towards lower energies from W-SAs to WC NPs, and are located between W foil and metallic $WO_2$ reference samples (Supplementary Fig. 14), indicating a reduced valence state of the tungsten species with metallic states[5], and this was also confirmed by Bader charge analysis (Supplementary Fig. 15). Correspondingly, as shown in the R-space plots of the FT-EXAFS profiles, the reference sample of W foil exhibits typical scattering path of W-W coordination at 2.85 Å, which is almost absent in that of W-SAs, suggesting the tungsten species are basically atomically dispersed. In contrast, W-W coordination is clear in W-ACs, indicating the existence of W-W bonds, while the intensity of the W-W peak is much weaker than that of WC NPs, implying the small size and unsaturated nature of W-ACs[29]. In addition, the observed predominant W-C/O (1–2 Å) scattering paths in W-ACs verify that they are rich in C/O ligands. To obtain the quantitative structural parameters of the coordination environment around the tungsten atoms, the W $L_3$-edge EXAFS spectra at the first shell (<2 Å) were curve fitted for W-SAs, W-ACs and WC NPs (Supplementary Fig. 16 and Supplementary Table 4). Meanwhile, the C K-edge near edge X-ray absorption fine structure (C K-edge NEXAFS) spectra of the as-obtained three tungsten species were also investigated in comparison with pure P-doped carbon substrates (Supplementary Fig. 17). All tungsten species and pure P-doped carbon substrates exhibit three prominent peaks at 285.5, 288.6, and 292.7 eV, corresponding to the C-C π*, C-O-C/C-N-C π* and C-C σ* transitions, respectively, where an obviously broadened C-C π* peak is observed in the

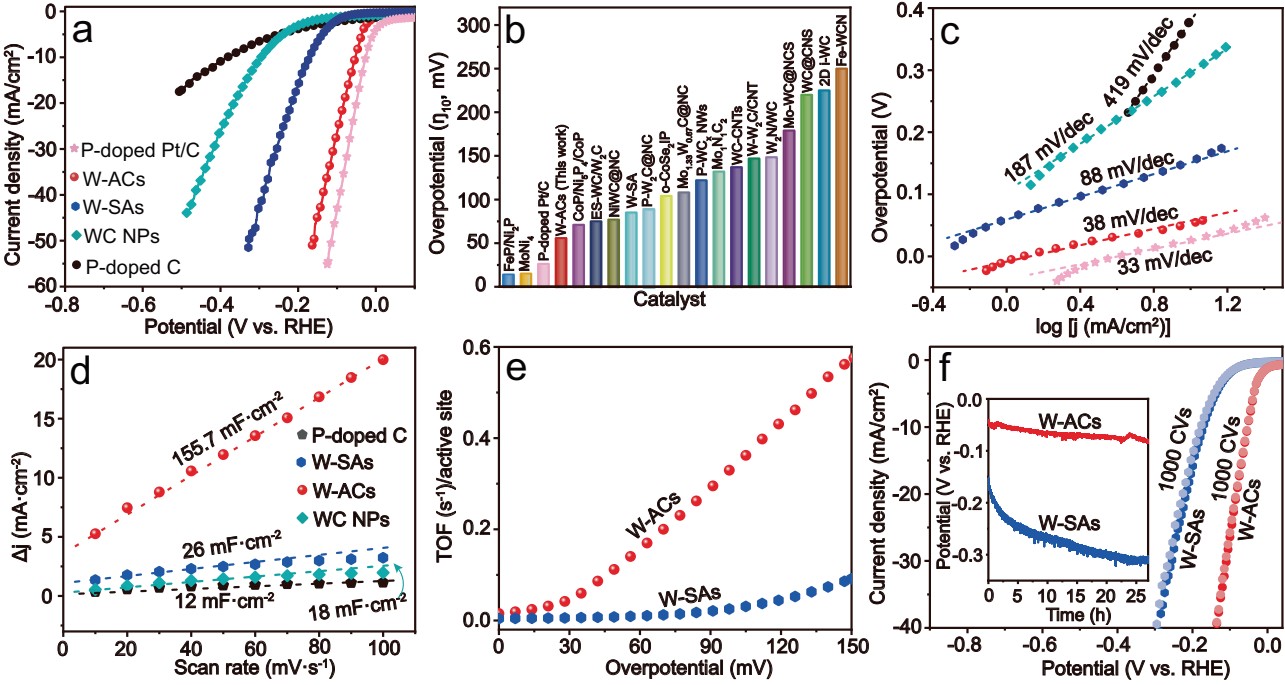

**Fig. 3 Evaluation of the alkaline HER performance using W-ACs in 1 M KOH solution. a** Polarization (LSV) curves and (**c**) Tafel plots for P-doped C, WC NPs, W-SAs, W-ACs, P-doped Pt/C catalysts. **b** The overpotentials at 10 mA/cm² extracted from W-ACs and P-doped Pt/C catalysts in this work, in comparison with those of non-noble metal catalysts previously reported in alkaline media[3,6,10,16,23,32–44]. **d** Electrochemical double-layer capacitance ($C_{dl}$) obtained from CV measurements between −0.8 and −0.6 V (vs. Ag/AgCl) at different scan rates (100–10 mV/s) for P-doped C, WC NPs, W-SAs and W-ACs. **e** TOF curves of W-SAs, and W-ACs. **f** LSV curves for W-SAs and W-ACs before and after 1000 CVs, with the chronopotentiometry measurements at current density of 10 mA/cm² in the inset.

tungsten-containing species, which is definitely different from the relatively sharp feature in pure P-doped carbon substrates, implying that a portion of the C 2p orbitals in C-C bonds are shared with the W 5d electrons via W-C bonds[30]. In addition, the role of the P dopant in three tungsten catalysts are also revealed by P 2p XPS measurements and Bader charge analysis. Only P-C (132.6 eV) and P-O (133.8 eV) bonds at higher binding energies are observed in P 2p core-level XPS spectra of W-SAs and W-ACs after deconvolution[31], while a small and broad P-W (129.1 eV) signal at lower binding energy is found for WC NPs[32] (Supplementary Fig. 18), indicating tungsten atoms in W-SAs and W-ACs are not directly coordinated with P atoms in the carbon matrix. Bader charge analysis shows that P-doping can regulate the oxidation states of the W atoms in W-W dual-atoms (Supplementary Fig. 15c–f). In particular, the W sites having a high oxidation state are beneficial for the adsorption of the produced OH* intermediates, whereas the lower valence counterparts serve as the active sites for the produced H* intermediates in alkaline HER electrocatalysis. Based on above comprehensive spectroscopic characterization and DFT calculations, the local atomic structures of the as-obtained various tungsten species are shown in Fig. 2e–g. The W-SAs are coordinated with three carbon atoms, W-ACs containing typical W-W bonds are anchored by carbon atoms in carbon substrates, and the larger P-doped WC NPs show hexagonal packing.

**Evaluation of alkaline HER electrocatalysis over the tungsten atomic clusters**. The HER performance of the as-obtained W-ACs was evaluated using a standard three-electrode system in 1 M KOH electrolyte. The pure P-doped C material, W-SAs, WC NPs, and a P-doped commercial 20 wt% Pt/C (P-doped Pt/C) reference catalyst were also examined for comparison (Supplementary

Figs. 19, 20). The linear sweep voltammetry (LSV) curves for all catalysts were presented in Fig. 3a, and all potentials measured against the Ag/AgCl electrode were calibrated with respect to the reversible hydrogen electrode (RHE) (Supplementary Fig. 21). The P-doped C sample exhibited very poor alkaline HER activity with an overpotential at 10 mA/cm² ($\eta_{10}$) up to 381 mV, whereas a remarkable enhancement in HER activity was observed for all tungsten-carbide-based materials, indicating the anchored tungsten species are the major contributor to the enhanced alkaline HER activity rather than the underlying P-doped carbon materials. As expected, among various tungsten-based materials, the W-ACs sample showed the best HER performance very close to that of commercial Pt/C. Specifically, an ultralow overpotential ($\eta_{10}$ = 53 mV) was achieved over the W-ACs, which was much lower than those of the WC NPs ($\eta_{10}$ = 298 mV) and W-SAs ($\eta_{10}$ = 171 mV), excelling most previously reported tungsten carbide-based materials[3,6,10,16,25,32–44], and even being competitive to the P-doped Pt/C ($\eta_{10}$ = 25 mV) (Fig. 3b and Supplementary Fig. 22). Additional kinetic information was extracted from the Tafel plots. The as-obtained W-ACs showed a small Tafel slope (38 mV/dec), which was much lower than those of P-doped (419 mV/dec), WC NPs (187 mV/dec) and W-SAs (88 mV/dec) (Fig. 3c), suggesting a fast alkaline HER electrocatalysis process[45]. The electrochemically active surface area (ECSA) is a significant parameter for revealing the intrinsic electrocatalytic activity of different catalysts and can be determined from the double-layer capacitance ($C_{dl}$) using cyclic voltammetry (CV) measurements (Supplementary Fig. 23). A larger $C_{dl}$ usually implies more active sites for the alkaline HER electrocatalysis. Compared to the well-dispersed isolated W sites in W-SAs (26 mF/cm²) and crystallized WC NPs (18 mF/cm²), an obvious increase in the $C_{dl}$ (155.7 mF/cm²) was obtained for the W-ACs sample (Fig. 3d), suggesting that the W-ACs possess

more available active sites for catalyzing the alkaline HER process due to the formation of W-W dual-atoms, which maximize the atomic utilization efficiency[46]. In addition, W-ACs also yield a high TOF value of 0.12 $H_2$ $s^{-1}$ at an overpotential of 50 mV, a nearly 15-fold enhancement in TOF compared with that of W-SAs, indicating the high efficiency of the active sites for $H_2$ production (Fig. 3e). Electrochemical impedance spectroscopy (EIS) was also performed to evaluate the charge transfer resistance of the W-ACs in alkaline HER electrocatalysis. The semicircle shown in Nyquist plots indicates the charge transfer resistance between the electrolyte and the working electrode for the reduction reaction of $H_2O$[47]. As shown in Supplementary Fig. 24, the charge transfer resistance for W-ACs catalyzed alkaline HER process was determined to be approximately 12 Ω, which is much smaller than those of P-doped C, WC NPs and W-SAs, indicating faster charge transfer kinetics for HER on W-ACs. In addition, the alkaline HER performance of W-ACs with and without P-doping were also examined in Supplementary Figs. 25, 31. Compared to the P-doped sample, W-ACs without P-doping showed inferior HER activity with a larger overpotential ($\eta_{10} = 140$ mV), Tafel slope (80 mV/dec) and charge transfer resistance, suggesting that P-doping plays a vital role in improving alkaline HER performance of W-W dual-atoms.

Another significant criterion affecting the practicality of alkaline HER catalysts is their long-term durability. The durability of the W-ACs was first evaluated by continuous potential cycling from 0 to −0.2 V at a scan rate of 50 mV/s in 1 M KOH electrolyte. As shown in Fig. 3f, the alkaline HER activity was well retained over W-ACs after 1000 CVs, whereas an obvious negative shift of approximately 8 mV at 10 mA/cm² was observed on W-SAs, indicative of activity loss. To demonstrate the stability of the W-AC catalyst, prolonged chronopotentiometry measurements were performed at a current density of 10 mA/cm² in 1 M KOH electrolyte, and an approximately 90% increase of the overpotential at 10 mA/cm² was observed for W-SAs after long-term alkaline HER electrocatalysis, which may be attributed to its weak oxidation resistance (Supplementary Fig. 26). In contrast, the as-obtained W-ACs maintained a relatively steady HER activity over 1 day (inset in Fig. 3f), highlighting the improved structural robustness of W-ACs in alkaline electrolyte when compared to the W-SA counterparts.

**Quasi in situ alkaline HER electrocatalysis on tungsten atomic clusters**. Typically, Pt-like HER behavior implies the fast desorption of H* and OH* intermediates, and for tungsten carbide catalysts, the desorption of the produced OH* intermediates is crucial to avoid the oxidation of W sites. Hence, quasi in situ alkaline HER measurements monitored by the analysis chamber of NAP-XPS system were performed to clarify the adsorption and desorption behaviors of OH* and H* intermediates for the W-ACs-catalyzed alkaline HER process (Supplementary Fig. 27). Specifically, the W 4f and O 1s core-level XPS spectra were used to probe the OH* intermediates on the surface of the W-ACs, whereas the change in the valence band of W-ACs was employed for H* intermediate monitoring[48]. As shown in Fig. 4a, W-ACs treated at different negative potentials versus RHE in a glove box were transferred to the analysis chamber for the XPS measurements through the vacuum channel. The applied negative potentials can result in the cleavage of H-OH bonds, and accelerate the desorption of OH* intermediates. Usually, the complete elimination of the formed OH* at lower potentials suggests the good desorption of OH* intermediates, which facilitates the adsorption of H* intermediates (Supplementary Fig. 28), and prevents OH-induced oxidation problems. The W 4f core-level XPS spectra of W-ACs soaked in KOH solution without

electrochemical treatment (initial state) presents inevitable W-O (W-O/W-C(O)) signals at high binding energies after deconvolution (Fig. 4b, Supplementary Table 5), which can be attributed to the adsorbed $OH^-$ and robust oxygen species. Interestingly, W-ACs treated at 0.00 V versus RHE exhibited an obviously enhanced concentration of W-O (W-O/W-C(O)) bonds compared with that of the sample treated only by soaking in KOH electrolyte (Supplementary Tables 6, 7), and the increased content should be attributed to the OH* intermediates rather than $OH^-$ anions in the electrolyte, which can be understood in the following two ways: (i) the W-AC-catalyzed alkaline HER process takes place at the cathode, which will hinder the adsorption of the $OH^-$ anions in electrolyte because of the Coulomb effect during the process of alkaline HER; (ii) a large amount of electrons are accumulated at the cathode during the process of alkaline HER, which will protect the W-AC catalyst from oxidation to higher valence state, implying the W-O signals detected on used W-ACs were not stronger than that of the initial state when the applied negative potentials were removed. Meanwhile, the sharp increase of W-O (W-O/W-C(O)) signals indicates that the produced OH* intermediates were mainly adsorbed on the W centers, and the cleavage of H-OH bonds can be achieved at low potentials using W-AC catalysts. Besides, the observed slightly positive shift of the binding energies in W-O signals evidenced the higher oxidation state of the tungsten oxides in comparison with that of the initial state (Supplementary Table 5). As expected, the intensities of signals corresponding to W-O bonds reduced when the negative potentials increased from −0.02 to −0.06 V. Specifically, the XPS spectra of the W-ACs treated at −0.06 V exhibited obviously decreased intensities of W-O (W-O/W-C(O)) signals (Supplementary Table 6), and even lower than that of initial state, suggesting that OH* could be easily detached from W centers into the electrolyte at low potentials, and no more $OH^-$ anions (electrolyte) were adsorbed on the catalyst surface in comparison with the initial state when the applied negative potentials were removed. Correspondingly, the O 1s core-level spectra of W-ACs treated at −0.06 V presented the lowest concentration of W-O bonds, indicating the OH* intermediates were removed (Fig. 4c, Supplementary Table 8). Besides, a similar change was also observed in the valence band of W-ACs treated at different potentials (Fig. 4d). The largest decrease in the density of states and down-shift of the valence band edge were found in W-AC sample treated at 0.00 V, because the catalyst surface was covered a large amount of adsorbed $H_2O^*$, H*, and OH* intermediates, which largely reduces the W 5d density of states. However, the valence band for the sample treated at −0.06 V recovered, and even increased beyond that of the initial state (black line), suggesting the adsorbed alkaline HER intermediates were efficiently detached from the catalyst surface at low potentials (Fig. 4d). In addition, in alkaline HER electrocatalysis, the electrons are consumed by the adsorbed hydrogen species ($H_2O^*$, H*) rather than OH*, and the detachment of OH* intermediates is mainly caused by the Coulomb effect according to the alkaline HER reaction mechanism (inset in Fig. 4d)[49]. Therefore, as compared to OH* intermediates, the Pt-like desorption behavior of H* intermediates caused by the fast charge transfer from W 5d orbital to H 1s orbital should be responsible for the recovery of the valence band in W-ACs catalyzed alkaline HER.

**DFT calculations**. To elucidate the intrinsic relationship between the electronic structure and the enhanced alkaline HER activity of the as-obtained W-ACs, comprehensive DFT calculations were performed to determine the kinetic energy barriers in prior water dissociation step (Volmer step, $\Delta G_{H_2O}$) and subsequent hydrogen desorption process (Tafel step, $\Delta G_H$) according to the as-built

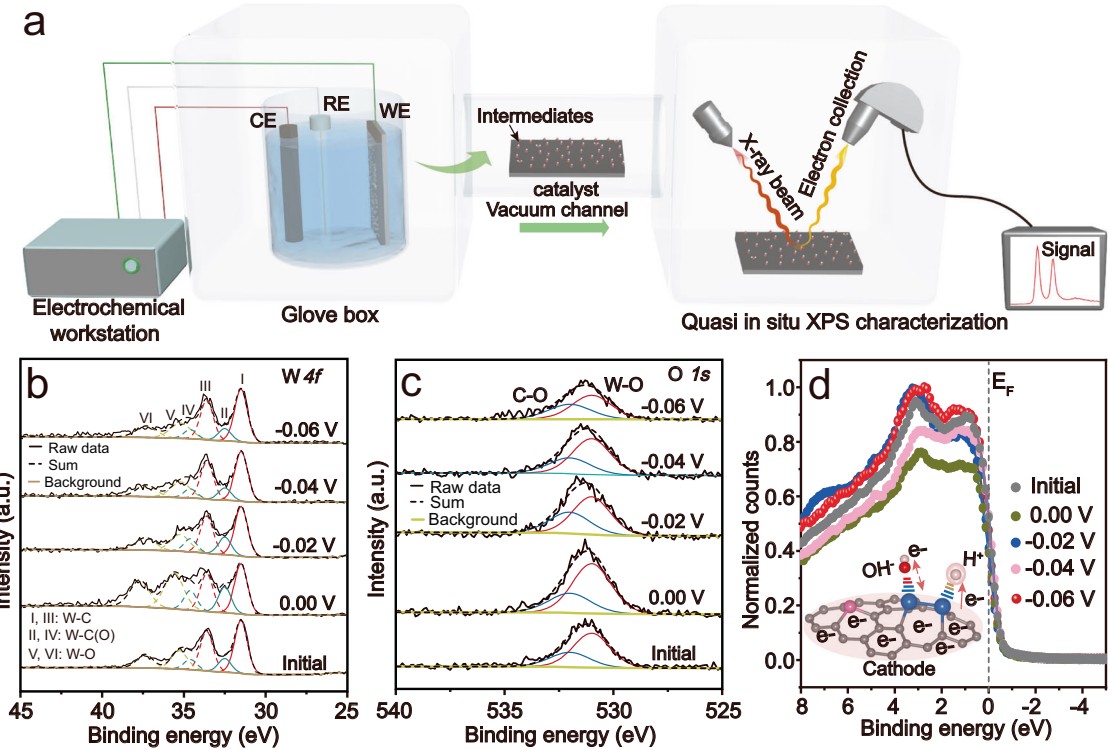

**Fig. 4 Quasi in situ alkaline HER measurements for W-ACs treated at different applied potentials. a** Schematic illustration of the quasi in situ alkaline HER experimental setup, where WE, RE, and CE represent the working, reference and counter electrodes, respectively. **b** W *4f* and (**c**) O *1s* core-level XPS spectra of W-ACs treated at 0.00, –0.02, –0.04, and –0.06 V versus RHE and soaked in KOH solution, respectively. **d** The corresponding valence band of above treated W-AC samples. Inset shows the typical desorption of H* and OH* intermediates from W-W dual-sites.

adsorption models of $H_2O$, activated $H_2O$, OH* and H* intermediates on W-SAs, W-ACs and the (001) facet of WC NPs, respectively (Fig. 5a–c and Supplementary Fig. 29)[39], where the DFT models were constructed based on the results of multiple characterizations (STEM, XPS, XANES, EXAFS). As shown in Fig. 5d, the benchmark Pt (111) catalyst exhibits the highest water dissociation energy barrier ($\Delta G_{H_2O}$, 0.81 eV) in comparison with three types of tungsten species, indicating noble Pt (111) catalyst is generally inefficient for the water dissociation step under alkaline conditions[50,51] (Supplementary Fig. 30). However, the activation barriers for $H_2O$ dissociation are remarkably reduced to 0.65 and 0.41 eV on W-SAs and WC NPs, respectively, indicating the single-atom or bulk carbide state is effective for the cleavage of H-OH bonds. Encouragingly, the value of $\Delta G_{H_2O}$ for the synthesized W-ACs further decreases to 0.24 eV, suggesting the sluggish Volmer step has been largely accelerated over the W-AC catalyst, which was further evidenced by the calculations based on a water-layer model containing metal cation[52,53] (Supplementary Fig. 32). After the formation of H* intermediates in prior water dissociation (Volmer) step, these species are desorbed from the active sites, thus yielding $H_2$ gas. Hence. the hydrogen adsorption free energy of H* intermediates ($\Delta G_H$) is a good activity descriptor for the hydrogen generation step (Fig. 5e). On the WC NP surface, the $\Delta G_H$ value is −0.61 eV, indicating a markedly strong binding affinity for H* intermediates, which prohibits the desorption of H* intermediates and blocks the active sites. For W-SAs, the value of $\Delta G_H$ is −0.46 eV, which is much smaller than that of WC NPs. Strikingly, the calculated $\Delta G_H$ value of W-ACs is efficiently decreased to −0.31 eV, which obviously surpasses those of W-SAs and WC NPs, and is even comparable to that of Pt (111) (−0.28 eV) (Supplementary Fig. 30g), suggesting the Pt-like behavior for the desorption of H* intermediates on W-AC catalyst. This

combined with the low energy barrier (0.24 eV) of water dissociation step, enables highly-efficient alkaline HER performance to be achieved on W-AC catalyst (Supplementary Fig. 33 and Supplementary Movie 1).

The enhanced performance of the synthesized W-ACs compared to those of W-SAs and WC NPs is also manifested by a comparison of the electronic interactions between the adsorbed H* and the tungsten-based catalysts. The calculation of the local density of states (LDOS) calculation is a powerful means to probe the interaction strength between W and H atoms[10], because a larger interaction energy usually implies stronger adsorption. As shown in Fig. 5f, the adsorbed H atom has a strong affinity for the nearest W atom on WC NPs, as indicated by the two significant hybridization energies centered at −7.7 and −5.6 eV, respectively, whereas a slightly rightward shift of the interaction positions (−3.8 and −1.9 eV) towards the Fermi level was observed on W-SAs, indicating a relatively decreased adsorption strength of W-H bonds on W-SAs. Encouragingly, W-ACs exhibit a favorable interaction strength for the W-H bonds with three hybridization peaks located at higher energy levels (about −3.1, −2.1, and −1.1 eV) in close to the Fermi level, suggesting the strong W-H interaction has been substantially weakened, which will facilitate the desorption of the H* intermediates from W-AC surface during the alkaline HER electrocatalysis.

In addition, the desorption of the OH* intermediates from these three tungsten species were also investigated by calculating the adsorption free energies ($\Delta G_{OH}$), which is closely related to the oxidation problem of W centers. As listed in Supplementary Table 9, the value of $\Delta G_{OH}$ for W-ACs is −1.42 eV, which is lower than that of the W-SAs (−1.65 eV), and much lower than those of previously reported excellent alkaline HER catalysts such as $MoNi_4$ (−4.89 eV)[39], NiFeRu-LDH (−4.86 eV)[54] and Ni-$MoS_2$

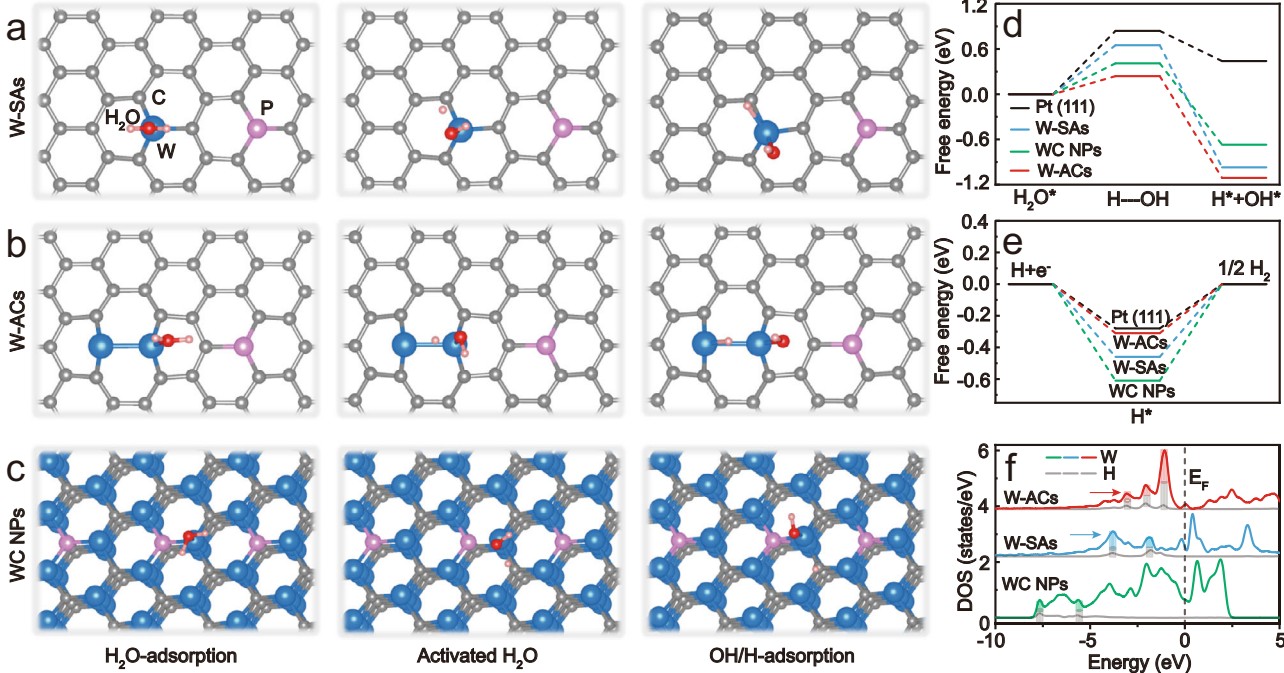

**Fig. 5 Proposed reaction mechanism of the alkaline HER electrocatalysis over the W-AC catalyst.** DFT calculated adsorption models for $H_2O$ and activated $H_2O$, $OH^-$, and $H^+$ on the surface of (**a**) W-SAs, (**b**) W-ACs and (**c**) WC NPs. Here, H, C, O, P, and W atoms are indicated by orange, gray, red, pink, and blue spheres, respectively. The corresponding calculated free energy diagrams for (**d**) water dissociation and (**e**) hydrogen desorption steps on the reference Pt (111), W-SAs, W-ACs, and WC NPs. **f** Local density of states for the W and H atoms on the surface of W-SAs, W-ACs, and WC NPs. The shaded marked positions indicate the interaction energy of W-H bonds.

$(-3.36 \, eV)$[55], indicating the formation of W-W bonds can weaken the adsorption free energy of OH* intermediates, which was also evidenced by the oxidative LSV measurements (Supplementary Fig. 34)[56]. Note that the OH* adsorption is still strong on W-ACs, and such a strong adsorption affinity will block the HER active sites. However, considering the fact that the alkaline HER process is driven by the increasing negative potentials, the detachment of OH* intermediates will be largely facilitated by the continuously enriched negative charge at the cathode, as determined by aforementioned quasi in situ alkaline HER measurements in Fig. 4.

## Discussion

In summary, by monitoring the evolution of the electronic structure of tungsten species from W-SAs to large-sized WC NPs using in situ variable-temperature NAP-XPS measurements, we confirmed that W-ACs with Pt-like electronic structure can be synthesized using W-SAs as the parent materials via a thermal migration strategy. Owing to the coordinately unsaturated W-W and W-C bonds, the energy barriers in prior water dissociation step and subsequent hydrogen desorption step are altered in alkaline HER electrocatalysis. Accordingly, the as-obtained W-ACs exhibit markedly high alkaline HER activity with an ultralow overpotential of 53 mV at 10 mA/cm$^2$ and a small Tafel slope (38 mV/dec), as well as a high TOF value of 0.12 s$^{-1}$ at an overpotential of 50 mV. In addition, our comprehensive quasi in situ HER measurements and DFT calculations reveal that the excellent alkaline HER activity of the W-ACs mainly originates from the accelerated desorption kinetics of H* and OH* intermediates on W-W dual-sites. These findings may provide a feasible route towards the rational design of atomic-cluster catalysts with high alkaline hydrogen evolution activity. On the basis of our findings, the further exploration of other Pt-like materials for highly-efficient catalysis is merited.

## Methods

**Chemicals**. Sodium tungsten dihydrate ($Na_2WO_4 \cdot 2H_2O$, 99.5% Sinopharm Chemical Reagent, Shanghai, China), polyethylene oxide-co-polypropylene oxide-co-polyethylene oxide (Tergitol Type XH (nonionic) aka $P_{123}$, 98%, Adamas Reagent, Shanghai), dopamine hydrochloride (DA 98% Aladdin, USA), and 2-amino-2-hydroxymethyl-propane-1,3-diol (Tris, 98%, Aladdin, USA).

**Preparation of tungsten precursors**. Tungsten precursors were synthesized in accordance with the following method: 3 g of polydopamine (PDA), 3 g of $P_{123}$, 0.5 g of Tris, 0.6 g of $Na_2H_2PO_4 \cdot 2H_2O$, and 0.04 g of $Na_2WO_4 \cdot 2H_2O$ were dissolved in distilled water (30 mL), then one piece of Ni foam was immersed into the mixture solution with continuous stirring for 6 h (Supplementary Fig. 1). As a result, the powder containing tungsten precursors and Ni foams coated with tungsten precursors were obtained by filtration, washed with a mixture of distilled water and ethanol, and then dried at 60 °C.

**Preparation of different tungsten-containing materials**. W-SAs were obtained by pyrolyzing the W precursor at 700 °C for 2 h in an Ar atmosphere (1 bar), followed by alkaline leaching in 6 M KOH solution for 3 days. The desired W-ACs were obtained by annealing the W-SAs at 550 °C for 30 min under an Ar/$H_2$ atmosphere (Ar: 300 sccm, $H_2$: 30 sccm, 1 bar). The large-sized WC NPs were collected by treating the W-SAs at a higher temperature of 650 °C for 30 min in an Ar/$H_2$ atmosphere (Supplementary Fig. 1).

**Characterization**. XRD patterns of the as-prepared tungsten-based nanomaterials were recorded on a Bruker AXS D8 Advance X-ray diffractometer with a Cu Ka radiation target (40 V, 40 A). The Raman spectra were recorded on a LabRAM HR Raman spectrometer (HORIBA Jobin Yvon) with an excitation laser of 532 nm. Standard TEM observation was performed using an FEI Tecnai G2F20 microscope. Atomic-level HAADF-STEM images and the corresponding STEM-EDS elemental maps were measured on an FEI Titan Themis Z 3.1 equipped with a SCOR spherical aberration corrector and a monochromator. The probe convergence angle was 80 mrad, and camera length was 115 mm in the STEM mode. The weight ratios of W and P elements in W-SAs and W-ACs were measured by the ICP-OES measurements (Thermofisher Avio 200).

**X-ray absorption spectroscopy**. C K-edge NEXAFS spectra were performed at the Catalysis and Surface Science End-station at the BL11U beamline in the National Synchrotron Radiation Laboratory (NSRL) in Hefei, China. The as-obtained tungsten-based powders were pressed on a conductive nickel adhesive (0.5 cm × 0.5 cm), and transferred to the ultrahigh vacuum (UHV) chamber for the

collection of the C K-edge NEXAFS spectra. The extended X-ray adsorption fine structure spectra (W $L_3$-edge) were collected at BL14W1 station at the Shanghai Synchrotron Radiation Facility (SSRF). For these measurements, the storage rings were operated at 3.5 GeV with a maximum current of 250 mA. A Si mono-chromator was used to collect the date of W $L_3$-edge data at room temperature, which was recorded in fluorescence excitation mode using a Lytle detector, and the spectrum of the standard W foil was recorded in transmission mode using an $N_2$-filled ionization chamber. The acquired EXAFS data were processed according to standard procedures using ATHENA software.

**In situ variable-temperature NAP-XPS measurement**. XPS measurements were performed using a SPECS NAP-XPS instrument with a temperature-controllable laser heating device and thermocouple equipped. The photon source is the monochromatic X-ray source of Al Kα (1486.6 eV), and the overall spectra resolution is Ag 3d5/2, < 0.5 eV FWHM at 20 kcps@UHV. The Ni foam coated with carbon-supported tungsten single atoms was fixed in the sample holder with a highly sensitive thermocouple. After pretreatment using Ar etching ($5 \times 10^{-6}$ mbar), the sample was transferred to the analysis chamber for the subsequent variable temperature XPS measurements. In a typical run, the blank XPS spectra including W 4 f and C 1 s core-level XPS spectra, as well as the corresponding valence band spectra were collected at 25 °C, followed by the heat-treatment of the tungsten-based sample using the laser at 400, 450, 500, 550, 580, or 600 °C in a pure hydrogen atmosphere (0.1 mbar) for 5 min. The above-mentioned data were recorded to monitor the evolution of the electronic structure of the W atoms as the W-SAs become large WC NPs. All binding energies were the raw XPS data without further calibration because the samples were sufficiently conductive to allow reliance on the calibration of the spectrometer alone.

**HER electrocatalysis under open conditions**. Electrochemical measurements were conducted on a CHI760E electrochemical station (Shanghai Chenhua Co., China) using a standard three-electrode system in 1 M KOH electrolyte, in which a glassy carbon (GC) electrode (diameter: 3 mm), Ag/AgCl electrode, and a carbon rod were used as working, reference, and counter electrodes, respectively. All potentials were collected against the Ag/AgCl electrode and calibrated with respect to a reversible hydrogen electrode (RHE) according to the calibration method in Supplementary Fig. 21, that is, $E_{RHE} = E_{Ag/AgCl} + 1.02$ V. In a typical synthesis of the catalyst ink, 5 mg of W-AC was dispersed in a mixture of 800 μL ethanol, 170 μL water, and 30 μL Nafion, followed by sonication for at least 30 min to form a homogeneous ink. Then, 10 μL of the catalyst ink was dropped onto the polished working electrode and dried naturally for further measurements.

**TOF calculations**. The $H_2$ conversion efficiency of W-SAs and W-ACs can be evaluated from the TOF value, which is obtained according to the following Eq. (1):

$$TOF(s^{-1}) = \frac{j}{2Fn} \quad (1)$$

where $j$ (A) is the current at a given overpotential, 2 is the number of electrons consumed to form 1 mol $H_2$, $F$ represents the Faraday constant (96500 C/mol), n (mol) is the number of moles of loaded W atoms on the GC electrode, which was determined by the ICP-OES analysis. The weight ratios of W in the W-SAs and W-ACs are 9.67 and 10.49 wt%, respectively. Accordingly, the TOF values of W-SAs and W-ACs at an overpotential of 50 mV are 0.008 and 0.12 s$^{-1}$, respectively. In addition, the weight ratio of P in W-SAs and W-ACs is 2.57 and 2.62 %, respectively.

**Quasi in-situ HER measurement**. A glove box connected to the NAP-XPS instrument through a vacuum channel was designed for the quasi in-situ HER mechanism investigation. W-ACs soaked only in 1 M KOH electrolyte were used as blank samples, and the samples treated at different overpotentials (0.00, −0.02, −0.04, and −0.06 V vs. RHE) were transferred to the analysis chamber for further XPS measurements.

**DFT calculations**. All DFT calculations were performed in the Vienna ab initio simulation package (VASP)[57]. The projector augmented wave (PAW) method was employed to describe the ion-electron interactions, and the generalized gradient approach (GGA) of the Perdew-Burke-Ernzerhof (PBE) functional was used to describe the electron-electron exchange correlation[58,59]. The van der Waals interaction was also considered by using the DFT-D3 dispersion correction throughout all calculations[60]. The cutoff energy for the plane-wave basis was set to 500 eV. The geometry optimization was converged when the maximum force on each atom reached to 0.03 eV/Å. The W-SA and W-AC models were constructed using a 6 × 6 supercell of graphene, and one or two C atoms were replaced by W atoms. A 4 × 4 × 1 k-points mesh was sampled in reciprocal space, and a vacuum space of 15 Å was introduced to avoid the interaction between the periodic images. The nature of the transition states was confirmed using the climbing image nudged elastic band (CI-NEB) method[61].

The adsorption free energy for hydrogen is defined as Eq. (2):

$$\Delta G = \Delta E_H + \Delta E_{ZPE} - T\Delta S \quad (2)$$

where the $\Delta E_H$ is the adsorption energy of the H atom on the substrate obtained from DFT calculations. The $\Delta E_{ZPE}$ and $\Delta S$ are the difference between the zero-

point energy and entropy between the adsorbed H and half of an $H_2$ molecule in the gas phase, respectively. The zero-point energy and entropy were calculated under standard conditions at 298.15 K.

## Data availability

Source data are provided with this paper, which can also be available from the corresponding authors on reasonable request. Additionally, data reported herein have been deposited in the Figshare database, and are accessible through https://doi.org/10.6084/m9.figshare.16780810. Source data are provided with this paper.

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

## Acknowledgements

This work was supported by the National Natural Science Foundation of China (No. 91845109, No. 21872169, No. 22109171, No. 22172190). Y.C. and J.M. would like to acknowledge the support from the CAS Project for Young Scientists in Basic Research (No. YSBR-022). Y.C. would like to acknowledge the support from the Young Cross Team Project of CAS (No. JCTD-2021-14). Z.C. would like to acknowledge the support from Jiangsu Planned Projects for Postdoctoral Research Funds (No. 2021K226B). The authors are grateful for the technical support (in situ NAP-XPS characterization) for Nano-X from Suzhou Institute of Nano-Tech and Nano-Bionics, Chinese Academy of Sciences (SINANO). We thank Q.X., J.H., H.D. and J.Z. for their help in Near Edge X-ray Absorption Fine Structure (C K-edge NEXAFS) performed at the Catalysis and Surface Science End-station at the BL11U beamline in the National Synchrotron Radiation Laboratory (NSRL) in Hefei, China. We thank B. M. and J.M. for their help in X-ray adsorption spectra (W-L₃ edge XAS) measurements at BL14W1 station in the Shanghai Synchrotron Radiation Facility (SSRF). We thank X.D for her help in the characterization and analysis of spherical aberration-corrected electron microscope.

## Author contributions

Y.C. and Z.Z. conceived the project and designed the experiments. Z.C., J.C., C.Z. and N.Y. performed the synthesis and characterization of catalysts and the electrocatalytic measurements. Z.C., D.D., H.L., W.W., X.G., Z.L. and Z.G. performed the in-situ variable temperature NAP-XPS measurements and quasi in-situ alkaline HER characterizations. Z.C. and X.D. performed the characterization of spherical aberration-corrected electron microscope. Z.C., J.M., R.G. and D.L. performed the X-ray adsorption experiments and analyzed the raw data. Z.C., Y.X. and L.W. carried out the DFT calculations. Z.C., G.S., S.C., L.W., Z.Z. and Y.C. wrote the paper. All authors discussed the results and commented on the paper.

## Competing interests

The authors declare no competing interests.
