## [Peer Review File · Nature Communications]

Thermal migration towards constructing W-W dual-sites for boosted alkaline hydrogen evolution reactionREVIEWER COMMENTS

Reviewer #1 (Remarks to the Author):

The manuscript "Significant Pt-like Alkaline Hydrogen Evolution Activity over Tungsten Atomic Clusters" reports on a new preparation route for a tungsten carbide catalyst material, which is probed with respect to its use for the hydrogen evolution reaction (HER) in alkaline media (and which entails electrochemical water splitting and dihydrogen formation). A very significant portion of the manuscript is concerned with the characterisation of the material and the preparation process. The following experimental techniques have been used for materials and process characterisation: XPS, STEM/TEM, XANES/NEXAFS spectroscopy, EXAFS spectroscopy and SEM. Supporting DFT calculations have been carried out that address atomic structure, energy, the density of states and ionisation states via a Bader charge analysis. The electrocatalytic function has been characterised by linear sweep and cyclic voltammetry, tafel plots (providing overpotentials and tafel slopes), TOFs and EIS. Hence, the work is based on a very comprehensive combination of methods that allows the authors to really develop a true understanding of the catalyst material.

In the manuscript, three different materials (derived in three different steps of the preparation process) are considered: single tungsten atoms (W-SAs), atomic tungsten clusters (W-ACs) and tungsten carbide nanoparticles (WC-NPs). The materials are supported on P-doped carbon. Comparison is made to a commercial Pt/C catalyst. The W-ACs – thought to be primarily W dimers (evidence is provided) – exhibit excellent activity for the HER as characterised by the required overpotential to reach an electrochemical current density of 10 mA/cm^2 and the tafel slope. I should make clear that I am not an electrochemist and that, therefore, my electrochemistry expertise is rather shallow. With this said, I find that an overpotential as low as 53 mV and a tafel slope as low as 38 mV/decade in the alkaline HER are really the outstanding results reported in the present work: I have not found any other report on tungsten carbide materials (which have been considered as a replacement for Pt since the 60s!) that even comes close to such low values, which, moreover, are very close those of Pt and therefore really justify the assumption that tungsten carbides, made from earth-abundant materials, may take the role of Pt in HER electrodes.

Both in the abstract and in the conclusions the authors stress the role of in situ monitoring by NAP-XPS (which I am an expert on as well as on other spectroscopic and microscopic characterisation). I do agree with the authors that their careful characterisation, using the wide range of methods listed above, is tremendously important and allows them to really take forward the preparation process and understanding of the characteristics of the material. I am a bit doubtful, however, about referring to the (XPS) characterisation as "in situ". For XPS to be called "near-ambient pressure x-ray photoelectron spectroscopy" (NAP-XPS) and to qualify as an in situ method it is necessary for the measurements to be carried out in the presence of a gas, vapour or even liquid phase in contact with the solid phase (the catalyst material) at a pressure of at least 10^{-3} mbar. Otherwise, it is just plain (ex situ) XPS – and there

is absolutely nothing wrong with this; most XPS characterisation can be and should be done using conventional, vacuum-based XPS. It is not clear what conditions have been used in the XPS measurements presented in the article: Were the XPS measurements during the formation of the W-ACs and WC-NPs carried out in the presence of the hydrogen atmosphere? The methods section can be read as if this was the case (i.e. at 0.1 mbar H₂) – but it is not clear. And if it was, was this really necessary, or would the same results have been obtained by heating the W-SAs at 550-580 °C in H₂ (or Ar/H₂), pumping down, transfer to the XPS chamber and measurement in vacuum? I strongly suspect that this would have been the case. Then using a NAP-XPS instrument and in situ monitoring would, indeed, be convenient since samples transfer can be avoided, but not necessary, which also implies that the same experiments and preparation could be done by many more research groups with access to a conventional XPS setup. Hence, I would like to ask the authors to make clear the conditions of the XPS measurements and whether in situ characterisation is "just" an additional benefit or if it is strictly necessary.

With regard to the in situ NAP-XPS question, I also note that the XPS data in figure 4 and in supplementary figures 5, 6 and 7 probably were carried out in vacuum. Then the measurements should be labelled "XPS" rather than "NAP-XPS" (and the conditions of measurements should be made clear in the methods section, figure captions or text). It should also be made clear what is meant by "quasi in situ" on page 14 of the manuscript.

As one can see from what I have written above, I think that the manuscript really implies a very important step forward towards being able to replace Pt electrodes in the HER in alkaline solution by tungsten carbide materials. I also think that the multi-method characterisation approach really has contributed to rendering this step forward possible and that, overall, the analysis of the characterisation data is thorough and well done. I do have a lot of comments to make, though, on detailed aspects. These comments are listed below, and I hope that they will help the authors to improve their manuscript.

1. It is clear from the methods section that both the tungsten atomic clusters and the tungsten carbide nanoparticles were formed by heating the single tungsten atoms under an Ar/H₂ atmosphere. This is not clear, and therefore slightly confusing, from the preparation process description on the third page of the main text. Since the preparation process is at the core of the manuscript, I would like to suggest that it is made clear in the main text that a hydrogen atmosphere is involved in the process.

2. In the methods section it is written that the W-ACs were obtained by annealing the W-SAs at 550 °C under an Ar/H₂ atmosphere. Does this imply that the total pressure was 1 bar (the question is valid for all statement of preparation in Ar or Ar/H₂ atmospheres)? Was there air present in the atmosphere?

3. The preparation route entails heating in an Ar/H₂ atmosphere. The XPS experiments entails heating in a hydrogen atmosphere. Was there any Ar present? If not, how did the authors ensure that the preparation process and the XPS experiment monitor the same chemical processes? Please comment also on the pressure difference of the preparation process and the XPS process. Furthermore, please see my comments above.

4. What were the experimental details of the XPS measurements? What kind of photon source (and hence which photon energy) was used? What was the overall spectral resolution? What was the pressure during the experiments? Since the samples mostly were carbon (plus small admixtures of tungsten carbide) and Ni foam, I guess the samples were metallic or at least semimetallic and that charging therefore was no problem. Could the authors please confirm and state in the manuscript?

5. Curve-fitting of the XPS data: Could the authors please specify what kind of lineshapes they used and which values they found for parameters such as peak position, gaussian width, lorentzian width etc., in all cases. It would be suitable if the parameters were provided in the supplementary information. I note that some lines look overly lorentzian in shape. Even background shapes and parameters should be stated.

6. The calibration of the photoelectron spectra were done "to the standard C 1s peak (284.6 eV)", i.e. to what by many authors is referred the "carbonaceous carbon peak". The problem is – there is nothing such as "carbonaceous carbon". What is adsorbed on a sample as "sample contamination" depends on the sample material, its history and the sample environment from which residual gases are released (and then adsorbed on the sample). There carbonaceous carbon peak is by no means a reliable reference. As I written above, I guess the samples were sufficiently conductive to allow reliance on the calibration of the spectrometer alone. For conductive samples the calibration of the spectrometer should be done thoroughly and in periodic intervals, but then one can rely on it.

7. It is written that the Ni foam used in the XPS experiment was coated with carbon-supported tungsten single atoms. Could the authors please provide more details on how they did this?

8. Supplementary Figure 6 and corresponding text in the manuscript: From the data shown here the intensities of the C 1s peak cannot be properly compared, since no proper background treatment/removal has been done. For the intensities to be comparable (at a glance) the backgrounds on the low- and high-binding energy sides must be the same for all spectra. In the case of the present spectra it would be advantageous to remove a polynomial or a Shirley background. With this said, a closer inspection indeed shows that the C 1s intensity decreases upon heating to 580 °C and increases again upon heating to 600 °C. With a proper data treatment this can be made visible in a much better way. With respect to the possible feature labelled "Carbide C": the signal-to-noise ratio does not really allow to make a definitive statement on its presence or absence. It might be there – or it might not. If

the authors really want to prove its (expected) existence, they will have to take a spectrum with significantly better statistics.

9. The authors curve-fitted the XPS data in Fig. 2b and, based on the curve-fitting results, they state that there is a shift to lower binding energy upon heating at 500 °C. The curve-fitting results are, however, not particularly good and a shift cannot really be inferred from them. A more reliable method would be to calculate the first moment of the W-C intensities (of either the 5/2 or the 7/2 peak or both, after background removal; a bit problematic is going to be the intensity of the W-O peak and of an unidentified structure on the high-binding energy sides of the W-C peaks). What is the unidentified structure? Could the authors please label the different lines?

10. Supplementary Figure 12: Quite generally the background treatment is not particularly good in any of the panels (the background is too low on the high-binding energy side). The background used in panel c is simply faulty, since it is far too low also on the low-binding energy side. It is essentially only due the poor background that a "P-W" component can be fitted. In order to really prove the existence of a P-W component, the authors will have to measure a spectrum with better statistics.

11. As is clear from my comments 8 and 10, I do not think that the XPS data can be used to prove the existence of a carbide for the WC-NPs or bond between P and W in the WC-NPs. This is not necessarily a major problem, given that the material amount is small. But, of course, XPS data with a sufficient signal-to-noise ratio where the corresponding features can be identified unambiguously would strengthen the case of the authors. Hence, the authors might want to consider measuring such XPS data.

12. It is difficult to link the colours of the spectra in Fig. 2c to those in the legend, partly because different line thicknesses have been used. The authors could think of a better way of identifying the different spectra. Similarly, the colour scale in Supplementary Figure 7 is chosen in such a way that it is difficult to distinguish the spectra of C@Ni foam and of W-SAs at 600 °C.

13. It is standard convention to display XPS data with a reverse binding energy scale, i.e. with high binding energies to the left and low to the right. The authors might want to consider following this convention in order to make it easier for the readers to consider the data.

14. Fig. 2d and Fig. 4d: The simultaneous use of green and red colours without further means of identification than a colour legend is not advisable, since around 5 % of the world population (or around 10 % of the male population) have a red/green colour vision deficiency.

15. Could the authors please render the following sentence on page 2 clearer: "Usually, unsaturated coordination occurs in tungsten carbides having diameters of less than 5 nm (or single tungsten atoms) owing to the defects by their ultrasmall size."

16. In Fig. 1 c a dashed red arrow is shown. In the text it is referred to as "the dashed orange arrow". Please correct.

17. Fig. 1f: The red arrows confused me a while until I understood that each of them indicates a structure that is identified with a W-AC. The caption text could be rendered clearer, e.g. "Enlarged STEM of a surface with W-ACs, which are indicated by red arrows." I am not quite sure that I understand how the W-ACs were identified. I can see quite much more structure in the image that looks the same to me. Could the authors please explain more thoroughly?

18. Fig. 1f: The inset is a bit funny in that it represents more or less the same magnification as the main image (the 5 Å length bar is around half the size of the 1 nm = 10 Å length bar). What is the meaning of the inset then? What is it that I am supposed to see and what extra information am I supposed to gain?

19. Fig. 1a, Supplementary Figures 9, 13, 19: Could the authors please identify the colours used for the different elements? (this has been partly done in some of the figures, but not in all, and where it has been done it is not easy to see)

20. The scales of Figs. 1b, e and h are very different scales, which makes it difficult to compare the morphologies. It would be better if images with identical sizes were compared. The comment also extends to the images in Supplementary Figures 3 and 4.

21. Supplementary Figures 9b and 13a are identical – therefore, I would like to recommend combining the two Supplementary Figures 9 and 13 into a single one.

22. Supplementary Figure 13: Even in a Bader charge analysis there is an uncertainty, since the derived Bader charges will e.g. depend on which functional was used in the DFT calculation. The differences in Bader charges on the tungsten atoms in the presence and absence of P in Supplementary Figure 13 are small. I would like to ask the authors to provide a qualified (and referenced) statement in the manuscript to which extent the difference in Bader charges is significant.

23. Supplementary Figure 18: The W 4f presented here for W-SAs before and after HER are helpful for assessing to which extent they are oxidised. It would be helpful if the same kind of data could be shown for the W-ACs and even the WC-NPs.

24. On page 9, the authors state that "the white-line intensities [in the XANES] spectra for all tungsten species are high than that of the W foil reference, suggesting that all tungsten species are in the oxidation state." Which oxidation state? (even 0, for a metallic atom, is an oxidation state!)

25. By convention, XANES spectroscopy in the soft x-ray regime is called either NEXAFS spectroscopy or x-ray absorption spectroscopy (XAS). The author might want to consider to adhere to the convention when it comes to the C K-edge NEXAFS spectra, which they call XANES spectra.

26. On page 10, referral is made to a "Table 1". This should be "Supplementary Table 1".

27. It is found that the doping with P plays a very significant role for the HER activity of the W-ACs (cf. Supplementary Figure 17). Hence, a proper comparison would be to P-doped Pt/C rather than standard commercial Pt/C. In the same vein, comparison should be made not to "bare C", but to P-doped "bare C" (Fig. 3a) (and the latter comparison is more important than the former to assess the activity of the W-ACs). I have not made a thorough literature search for such systems and their activity, but a quick search brought directly up a report on an increased activity of a P (and N) co-doped Pt/C material, albeit in acidic medium (Wang et al., Nano Research 10 (2017) 238).

28. Fig. 3b lacks references. I guess the data stem from the works cited on line 10 of page 12?

29. I guess "JCPDF" on page 5 should read "JCPDS" (which today actually is the ICDD).

Reviewer #2 (Remarks to the Author):

This work has developed a W atomic cluster (W-ACs) supported on P-doped carbon materials as the electrocatalysts of alkaline HER. The coordination environment and electronic structure of W-AC have been characterized. A high HER activity close to commercial Pt/C catalysts was obtained on W-ACs. DFT calculations were performed to rationalize the high activity of W-AC. However, there are some critical questions in theoretical calculations.

1. As known, the structure of metal cluster is often not well defined. In DFT calculations, why do you select a double-atom model for W-ACs, namely two adjacent W atoms anchored on the graphene substrate? Typically, metal cluster can contain dozens of atoms, hence a double-atom model is not reasonable.

2. In experiments, Pt/C exhibits a higher HER activity than W-ACs (Figure 3a~c), while W-ACs shows much lower water dissociation barrier than Pt (Fig. 5d), which are inconsistent. Although H* desorption on Pt is a little easier than on W-ACs, water dissociation is the rate-determining step on Pt, with a barrier of 0.81 eV.

3. In DFT calculations, the adsorbed H₂O was considered as the initial state (IS) for water dissociation process. However, H₂O in solution can dissociate into adsorbed H* and OH- directly, namely $\text{H}_2\text{O}(\text{l}) \rightarrow \text{H}^* + \text{OH}(\text{l})$, which is more reasonable than the adsorbed H₂O dissociation ($\text{H}_2\text{O}^* \rightarrow \text{H}^* + \text{OH}^*$) in alkaline HER.

4. Why P-doped W-ACs shows better activity than the W-ACs without P-doping? What the role of P-doping? Although the authors have analyzed Bader charge (Fig. S13), the free energy diagram of HER on W-ACs without P-doping is also needed to compare with the P-doped sample.

5. For molecule H₂ formation from adsorbed H*, the energy barrier (activation energy) should be calculated to evaluate the reaction kinetics.

6. How did you calculate the adsorption free energy of OH*? Was the adsorption free energy referenced to hydroxyl (OH-) in solution, or neutral OH in vacuum (namely DFT calculated OH energy), or water? The OH* adsorption energy on W-ACs reached to -1.42 eV, indicating the desorption of OH* is very difficult on ambient condition, blocking the HER.

In light of the above questions, I don't recommend the publication of this manuscript in Nature Communications.

Reviewer #3 (Remarks to the Author):

The manuscript reported a tungsten atomic cluster (W-AC) catalyst for efficient alkaline HER. The authors apply a novel strategy to control the scale of tungsten atoms by thermal migration and prove that tungsten atoms could migrate and be aggregated into clusters at specific temperature (above 550 °C) via in-situ NAP-XPS. This work is quite impressive and it seems that these operando characterization experiments were carefully designed despite the difficulty of X-ray based operando analysis. Along with several experimental clues, DFT simulation also gives an adequate explanation for why 'W-ACs' facilitate alkaline HER catalysis. I have several questions and comments as follows.

More specific comments:

1. From the FT-EXAFS spectra at W L3-edge (Fig. 2d), the W-C/O scattering path was assigned around 1.5~2 Å. However, W-C scattering path in the tungsten carbide is usually placed about 2.1 Å (Nat Commun 2016, 7, 13216, The author already cited). The coordination of W-substrate carbon and W-C coordination in tungsten carbide should be distinguished.
2. From the W L3-edge XANES (Figure S8), not only the pre-edge position, the white line intensity of WC-NP should be also considered to evaluate the valence states. The author mentioned that WC-NP is more reduced than W-SA and W-AC due to the lower edge shift, but white line intensity of WC-NP is larger than those of them, suggesting that the tungsten in WC-NP is more oxidized.
3. In the constructed W-SA and W-AC models for DFT calculation, both models only contain one P atom which is replaced with C atom in a 6×6 supercell of graphene. A P atom in carbon substrate also might give an electronic effect on the binding energy of HER intermediate on the W active site, and the Bader charge of an adjacent tungsten atom can be calculated differently depending on the number of doped P atoms. Is there any specific reason for this composition?
4. In Fig 2c, the XPS valence band spectrum of W4f at 580 °C is highly closer to the Pt reference sample compared to that of W4f at 550 °C. But, the optimal temperature of 'W-ACs' synthesis is 550 °C in Methods. Why do authors set the synthesizing temperature at 550 °C for 'W-ACs'? Is there any catalytic performance results using W-based single atom electrocatalyst treated in 580 °C ?
5. There are some typos or errors in the manuscript.

Line 21 in page 13, "relatively steady HER activity over 1 day"

Line 26 in page 18, "(Supplementary Fig. 9c)"

Figure 3 a has mis-labeled W-SAs with wrong color in legend (Blue -> Orange)

Answer to reviewer's comments:

Replies on comments of reviewer 1:

The manuscript "Significant Pt-like Alkaline Hydrogen Evolution Activity over Tungsten Atomic Clusters" reports on a new preparation route for a tungsten carbide catalyst material, which is probed with respect to its use for the hydrogen evolution reaction (HER) in alkaline media (and which entails electrochemical water splitting and dihydrogen formation). A very significant portion of the manuscript is concerned with the characterisation of the material and the preparation process. The following experimental techniques have been used for materials and process characterisation: XPS, STEM/TEM, XANES/NEXAFS spectroscopy, EXAFS spectroscopy and SEM. Supporting DFT calculations have been carried out that address atomic structure, energy, the density of states and ionisation states via a Bader charge analysis. The electrocatalytic function has been characterised by linear sweep and cyclic voltammetry, tafel plots (providing overpotentials and tafel slopes), TOFs and EIS. Hence, the work is based on a very comprehensive combination of methods that allows the authors to really develop a true understanding of the catalyst material.

In the manuscript, three different materials (derived in three different steps of the preparation process) are considered: single tungsten atoms (W-SAs), atomic tungsten clusters (W-ACs) and tungsten carbide nanoparticles (WC-NPs). The materials are supported on P-doped carbon. Comparison is made to a commercial Pt/C catalyst. The W-ACs-thought to be primarily W dimers (evidence is provided)-exhibit excellent activity for the HER as characterised by the required overpotential to reach an electrochemical current density of 10 mA/cm^2 and the tafel slope. I should make clear that I am not an electrochemist and that, therefore, my electrochemistry expertise is rather shallow. With this said, I find that an overpotential as low as 53 mV and a tafel slope as low as 38 mV/decade in the alkaline HER are really the outstanding results reported in the present work: I have not found any other report on tungsten carbide materials (which have been considered as a replacement for Pt since the 60s!) that even comes close to such low values, which, moreover, are very close those of Pt and therefore really justify the assumption that tungsten carbides, made from earth-abundant materials, may take the role of Pt in HER electrodes.

Both in the abstract and in the conclusions the authors stress the role of in situ monitoring by

NAP-XPS (which I am an expert on as well as on other spectroscopic and microscopic characterisation). I do agree with the authors that their careful characterisation, using the wide range of methods listed above, is tremendously important and allows them to really take forward the preparation process and understanding of the characteristics of the material. I am a bit doubtful, however, about referring to the (XPS) characterisation as "in situ". For XPS to be called "near-ambient pressure X-ray photoelectron spectroscopy" (NAP-XPS) and to qualify as an in situ method it is necessary for the measurements to be carried out in the presence of a gas, vapour or even liquid phase in contact with the solid phase (the catalyst material) at a pressure of at least 10^{-3} mbar. Otherwise, it is just plain (ex situ) XPS - and there is absolutely nothing wrong with this; most XPS characterisation can be and should be done using conventional, vacuum-based XPS. It is not clear what conditions have been used in the XPS measurements presented in the article: Were the XPS measurements during the formation of the W-ACs and WC-NPs carried out in the presence of the hydrogen atmosphere? The methods section can be read as if this was the case (i.e. at 0.1 mbar H_2) - but it is not clear. And if it was, was this really necessary, or would the same results have been obtained by heating the W-SAs at 550-580 °C in H_2 (or Ar/ H_2), pumping down, transfer to the XPS chamber and measurement in vacuum? I strongly suspect that this would have been the case. Then using a NAP-XPS instrument and in situ monitoring would, indeed, be convenient since samples transfer can be avoided, but not necessary, which also implies that the same experiments and preparation could be done by many more research groups with access to a conventional XPS setup. Hence, I would like to ask the authors to make clear the conditions of the XPS measurements and whether in situ characterisation is "just" an additional benefit or if it is strictly necessary.

With regard to the in situ NAP-XPS question, I also note that the XPS data in figure 4 and in supplementary figures 5, 6 and 7 probably were carried out in vacuum. Then the measurements should be labelled "XPS" rather than "NAP-XPS" (and the conditions of measurements should be made clear in the methods section, figure captions or text). It should also be made clear what is meant by "quasi in situ" on page 14 of the manuscript.

As one can see from what I have written above, I think that the manuscript really implies a very important step forward towards being able to replace Pt electrodes in the HER in alkaline solution by tungsten carbide materials. I also think that the multi-method characterisation approach really

has contributed to rendering this step forward possible and that, overall, the analysis of the characterisation data is thorough and well done. I do have a lot of comments to make, though, on detailed aspects. These comments are listed below, and I hope that they will help the authors to improve their manuscript.

Author reply: Thanks for the referee's careful reading and detailed comments. We have made a point-by-point response to address the referee's concerns comprehensively and accurately, which contains the questions declared in above comments (Labeled as Q1, Q2, and Q3) and the following separate comments (Labeled as comments 1, 2, 3.....).

Q1. I am a bit doubtful, however, about referring to the (XPS) characterisation as "in situ". For XPS to be called "near-ambient pressure X-ray photoelectron spectroscopy" (NAP-XPS) and to qualify as an in situ method it is necessary for the measurements to be carried out in the presence of a gas, vapour or even liquid phase in contact with the solid phase (the catalyst material) at a pressure of at least 10^{-3} mbar. Otherwise, it is just plain (ex situ) XPS - and there is absolutely nothing wrong with this; most XPS characterisation can be and should be done using conventional, vacuum-based XPS.

Author reply: Thanks for the referee's valuable comments. Two XPS measurements were performed in this work: (i) The observation of the structural evolution of tungsten species from W-SAs to WC NPs by thermal migration (structure characterization, Fig. 4b and c); (ii) the characterization of W-ACs after treatment with increasing negative potentials (HER mechanism investigation, Fig. 4). On the one hand, the structure characterization was carried out in the analysis chamber of NAP-XPS (SPEC NAP-XPS) under pure H₂ atmosphere at a pressure of 0.1 mbar, and the treating temperature was controlled by the laser heating device and the thermocouple equipped (Supplementary Fig. 8). Obviously, it is an in situ NAP-XPS measurement. On the other hand, the HER mechanism investigation was measured by the vacuum-connected system including glove-box and NAP-XPS (Fig. 4a, Supplementary Fig. 27), in which W-ACs treated by different potentials were transferred to the analysis chamber of NAP-XPS via vacuum channels, and then the W 4f and O 1s core level XPS signals, as well as the valence band spectrum were collected for comparison (Supplementary Fig. 27, Fig. 4a). Similar characterizing method named as "quasi in situ" investigation was also reported by Oswald et al., which demonstrated the intercalation mechanisms of the Li-ion battery using the transport chamber connecting glove box

and XPS system (*Anal. Bioanal. Chem.* **393**, 1871-1877 (2009)). Therefore, regarding the potential-treated process and the following XPS measurements, the whole process can be regarded as quasi in situ alkaline HER mechanism investigation. In addition, two advantages should be emphasized by using the vacuum-connected glove box and SPEC NAP-XPS system rather than the conventional ultra-high vacuum (UHV) XPS for the characterization of the electrochemically treated W-ACs: (i) compared to the operation of alkaline HER process in open air condition, followed by examination using UHV XPS, our work conducted in the vacuum-connected SPEC NAP-XPS system can avoid the air interference (e.g., O₂ adsorption) and any other contaminations, which was helpful in revealing the intrinsic XPS signals induced by electrocatalysis; (ii) the spent samples extracted from the KOH electrolyte can be timely characterized by the SPEC NAP-XPS system, for a immediate record of the adsorbed OH* intermediates and the corresponding signals of the structural change, which may otherwise be weakened or vanished in conventional UHV XPS, since the spent samples are required to reach a high vacuum (<10⁻⁵ Pa) in sample chamber (Vacuum pumping over 1 day) before transferring to the analysis chamber for XPS characterization. The relevant discussion has been added in the revised manuscript.

Supplementary Figure 27. (a) Schematic illustration of the in situ variable temperature NAP-XPS setup. (b) Photo of the vacuum-connected glove box and NAP-XPS system. (c) Schematic illustration of the quasi in situ alkaline HER experimental setup, where WE, RE, and CE represent the working, reference and counter electrodes, respectively.

Q2. It is not clear what conditions have been used in the XPS measurements presented in the article: Were the XPS measurements during the formation of the W-ACs and WC-NPs carried out in the presence of the hydrogen atmosphere? The methods section can be read as if this was the case (i.e. at 0.1 mbar H₂) - but it is not clear. And if it was, was this really necessary, or would the same results have been obtained by heating the W-SAs at 550-580 °C in H₂ (or Ar/H₂), pumping down, transfer to the XPS chamber and measurement in vacuum? I strongly suspect that this would have been the case. Then using a NAP-XPS instrument and in situ monitoring would, indeed, be convenient since samples transfer can be avoided, but not necessary, which also implies that the same experiments and preparation could be done by many more research groups with access to a conventional XPS setup. Hence, I would like to ask the authors to make clear the conditions of the XPS measurements and whether in situ characterisation is "just" an additional benefit or if it is strictly necessary.

Author reply: Thanks for the referee's valuable comments. Yes, the XPS measurements (Structure characterization) during the formation of the W-ACs and WC-NPs were carried out in the presence of the hydrogen atmosphere (0.1 mbar). In order to address the referee's concern on whether the same results can be obtained by altered processing, we have also treated the W-SA samples at 550 and 580 °C respectively in vacuum-connected CVD tubular furnace at H₂ atmosphere, followed by transferring the treated samples to the analysis chamber for XPS measurements via vacuum channels (Supplementary Fig. 11a). As can be seen, the two samples (550 and 580 °C) exhibited relatively weak W-C signals (Supplementary Fig. 11b), suggesting the surface carbide structure was probably affected by the diffused graphitic impurities and small amount of stubborn oxide atoms for the cooled down sample (*Angew. Chem. Int. Ed.* **53**, 5131-5136 (2014), *ACS Catal.* **2**, 765-769 (2012)). However, an obviously enhanced intensity of W-C peak was detected after alkaline treatment in the vacuum-connected glove box, in which the W-C carbide signals were extremely similar to those collected by in situ variable-temperature NAP-XPS measurements under the same temperature (Supplementary Fig. 11c), indicating the surface impurities (noncarbide carbon, oxygen species) affecting the detection of W-C carbide signals have been removed by alkaline treatment (*Angew. Chem. Int. Ed.* **53**, 5131-5136 (2014), *ACS Catal.* **2**, 765-769 (2012)). Moreover, the corresponding valence band at the Fermi level (0~2 eV) were also comparable to that observed by in situ variable-temperature NAP-XPS

measurements (Supplementary Fig. 11d), demonstrating their Pt-like electronic structure after alkaline treatment. Therefore, comparing the above results obtained in the vacuum-connected CVD tubular furnace and those of in situ variable temperature NAP-XPS measurements, two important findings should be pointed out: (i) in situ variable temperature NAP-XPS measurements can be used to directly detect the intrinsic signals of the aggregated tungsten species without any interference due to the protection from reducing atmosphere at high temperature; (ii) the vacuum-connected CVD and NAP-XPS system can clearly reveal the enhanced W-C carbide signals in relevance to the aggregation of W-SAs, while an additional surface cleaning procedure using alkaline treatment should be introduced in glove box to avoid interferences from the surface graphitic carbon and stubborn oxygen species for the cooled down sample, which is obviously more complicated than the in situ variable-temperature NAP-XPS measurements.

Supplementary Figure 11. (a) Schematic illustration of the vacuum-connected CVD-glove box-NAP-XPS setup. (b) W 4f core level XPS spectra of W-SAs annealed at 550 and 580 °C, and second-treatment by 1 M KOH solution, respectively. (c) Comparison of (c) the W 4f XPS spectra and (d) valence band after thermal (550 and 580 °C) and alkaline treatment, and previously collected signals by in situ variable-temperature NAP-XPS measurements.

Q3. With regard to the in situ NAP-XPS question, I also note that the XPS data in figure 4 and in supplementary figures 5, 6 and 7 probably were carried out in vacuum. Then the measurements should be labelled "XPS" rather than "NAP-XPS" (and the conditions of measurements should be made clear in the methods section, figure captions or text). It should also be made clear what is meant by "quasi in situ" on page 14 of the manuscript.

Author reply: Thanks for the referee's valuable comments. As mentioned in Author reply (Q1), the XPS data of the samples treated by treatments under increasing negative potentials in Figure 4 were collected in the analysis chamber of NAP-XPS system, and the whole process can be regarded as quasi in situ alkaline HER mechanism investigation, which can be labeled as XPS measurements. The Ar-etching treatment was proceeded in the analysis chamber of NAP-XPS system under Ar atmosphere (5×10^{-6} mbar), and the W 4f XPS signals were then recorded to determine the optimized Ar-etching time under vacuum condition (5×10^{-9} mbar, Supplementary Fig. 9), which can be labeled as XPS measurement. The W 4f, C 1s, and the corresponding valence band were collected in the analysis chamber of NAP-XPS system at the H₂ atmosphere (0.1 mbar) when the temperature increased from 400 to 600 °C (Fig. 2b and c, Supplementary Fig. 12), which can be labeled as in situ NAP-XPS measurements. For consistency, the valence band of the reference samples (Bare Ni foam and P doped C@Ni foam) were also obtained in the analysis chamber of NAP-XPS system (Supplementary Fig. 13), and the valence band profiles of tungsten species at 550 and 580 °C were directly extracted from Fig. 2c, which can be labeled as XPS measurements. The relevant conditions of measurements have been made clear in the method sections, figure captions and text.

Comments 1:

It is clear from the methods section that both the tungsten atomic clusters and the tungsten carbide nanoparticles were formed by heating the single tungsten atoms under an Ar/H₂ atmosphere. This is not clear, and therefore slightly confusing, from the preparation process description on the third page of the main text. Since the preparation process is at the core of the manuscript, I would like to suggest that it is made clear in the main text that a hydrogen atmosphere is involved in the process.

Author reply: Thanks for the referee's valuable comments. The scale-up synthesis of W-AC and WC powders were carried out in conventional CVD tubular furnace at 550/650 °C for 30 min under an Ar/H₂ atmosphere, respectively. The in situ variable-temperature NAP-XPS (structure characterization) was performed under pure H₂ atmosphere (0.1 mbar). The detailed conditions for preparation and characterization conditions involving hydrogen atmosphere have been added in the revised manuscript. In addition, three groups of control experiments were conducted for illustrating the consistency of the scale-up synthesis and structure characterization processes: (i) in situ variable-temperature NAP-XPS was conducted under Ar/H₂ atmosphere (0.1 mbar), in which the ratio of Ar/H₂ was of 10 : 1 by regulating the partial pressure; (ii) in situ variable-temperature NAP-XPS was conducted under pure Ar atmosphere (0.1 mbar); (iii) the W-AC powders were synthesized in conventional CVD tubular furnace at 550 °C under Ar/H₂ atmosphere with a pressure of 0.1 mbar, in which the low pressure was achieved by a high precision vacuum pump. A detailed discussion about the relevant results has been presented in comments 3.

Comments 2:

In the methods section it is written that the W-ACs were obtained by annealing the W-SAs at 550 °C under an Ar/H₂ atmosphere. Does this imply that the total pressure was 1 bar (the question is valid for all statement of preparation in Ar or Ar/H₂ atmospheres)? Was there air present in the atmosphere?

Author reply: Thanks for the referee's valuable comments. Yes, the total pressure of synthesizing W-AC powders was 1 bar, and the atmosphere was the mixture of Ar and H₂ in avoidance of air presence.

Comments 3:

The preparation route entails heating in an Ar/H₂ atmosphere. The XPS experiments entails heating in a hydrogen atmosphere. Was there any Ar present? If not, how did the authors ensure that the preparation process and the XPS experiment monitor the same chemical processes? Please comment also on the pressure difference of the preparation process and the XPS process. Furthermore, please see my comments above.

Author reply: Thanks for the referee's valuable comments. The XPS experiments entailed in a hydrogen atmosphere was pure H₂ atmosphere without Ar gas. In order to address the referee's

concern on how to ensure that the preparation process and the XPS experiment monitor the same chemical processes, additional experimental data have been provided in the revised manuscript. Firstly, in situ variable-temperature NAP-XPS experiments at Ar/H₂ (0.1 mbar) and pure Ar (0.1 mbar) atmosphere have also been conducted, respectively. For the W 4f XPS signals of the tungsten species collected above 500 °C at Ar/H₂ (0.1 mbar) atmosphere (Supplementary Fig. 10a), the W 4f_{7/2}(W-C) intensity increased upon heating to 550 °C and decreased when further heated to 580 °C, in which the enhanced intensity of the W-C bond indicated the formation of a high density of tungsten clusters, whereas the simultaneous coating of the graphitic carbon on the formed tungsten clusters should be responsible for the decreased intensity (*Angew. Chem. Int. Ed.* **53**, 5131-5136 (2014); *Appl. Catal. B: Environ.*, **203**, 684-691 (2017)). Similar change of the W 4f XPS signals has been observed under pure H₂ atmosphere (0.1 mbar), as depicted in the main text. Meanwhile, a small difference between these two experimental results should be pointed out: an obvious decrease of the W-C peak together with the slightly increased W-O signals were observed in the tungsten species treated below 500 °C under Ar/H₂ atmosphere (which was invisible under a pure H₂ atmosphere), indicating a partial oxidation of the tungsten species below 500 °C, probably originated from the chemisorbed oxygen in the surficial region of carbon substrate that released in the Ar/H₂ atmosphere. In contrast, a clear oxidation phenomenon with markedly strong W-O signals was observed in tungsten species from 400 to 600 °C under pure Ar atmosphere (Supplementary Fig. 10b), indicating the tungsten species have bonded to the substrate oxygen atoms without the protection of hydrogen atmosphere, which severely impeded the detection of the intrinsic W-C carbide structures. In comparison with the previous results obtained in pure H₂, the above two control experiments confirm the reducing atmosphere provided by H₂ gas plays a vital role in observing the structure evolution of tungsten species and identifying the optimized synthesized temperature of W-ACs. Secondly, the scale-up synthesis of W-AC powders was also carried out at 550 °C under Ar/H₂ atmosphere with a pressure of 0.1 mbar. As shown in Supplementary Fig. 20a, high density of W-ACs were observed in the HAADF-STEM image of tungsten species obtained at 550 °C. The corresponding HER activity (polarization curve, Tafel plot, Nyquist plot) was also evaluated in 1 M KOH electrolyte (Supplementary Fig. 20b-d). The value of overpotential (η_{10}), Tafel slope, and charge transfer resistance (R_{ct}) was determined to be as low as 49 mV, 38 mV/dec, and 9 Ω , respectively (Supplementary Fig. 20b, c and d), evidencing

the W-AC powders synthesized at Ar/H₂ atmosphere with a low pressure of 0.1 mbar can also achieve a high alkaline HER activity. Meanwhile, we would like to point out a small detail on the synthesis of W-AC powders in conventional CVD tubular furnace, in which the reducing atmosphere is usually provided by the mixture of Ar/H₂ atmosphere rather than pure H₂ gas, because the usage of pure H₂ gas should be avoided in our synthesis regarding its severe explosion in case of leakage accident.

Supplementary Figure 10. W 4f core level XPS spectra for tungsten species treated in the temperature range of 400 - 600 °C under (a) Ar/H₂ and (b) pure Ar atmosphere, respectively. (c) HAADF-STEM image of W-ACs synthesized in conventional CVD tubular furnace under Ar/H₂ atmosphere with a pressure of 0.1 mbar. (d) Polarization curves, (e) Tafel plots, and (f) Nyquist plots of W-ACs obtained under Ar/H₂ atmosphere at a pressure of 0.1 mbar and 1 bar, respectively.

Comments 4:

What were the experimental details of the XPS measurements? What kind of photon source (and hence which photon energy) was used? What was the overall spectral resolution? What was the pressure during the experiments? Since the samples mostly were carbon (plus small admixtures of tungsten carbide) and Ni foam, I guess the samples were metallic or at least semimetallic and that charging therefore was no problem. Could the authors please confirm and state in the manuscript?

Author reply: Thanks for the referee's valuable comments. As depicted in manuscript and above comments, two important XPS measurements were performed in this work, one is the in situ variable-temperature NAP-XPS measurement (structure characterization), the other is the quasi in

situ alkaline HER mechanism investigation. For the structure characterization, the sample of W-SAs coated Ni foam was initially treated by Ar-etching under vacuum conditions to remove any possible remnants of the surface WO_x passivation layer and graphitic carbon without impacting the W-C carbide structure (Supplementary Fig. 9). Subsequently, the treated sample was transferred to the analysis chamber for variable-temperature NAP-XPS characterization, and the W 4f/C1s core level XPS signals and the corresponding valence band spectrum were in situ recorded with the increase of temperature from 400 to 600 °C under H_2 atmosphere (0.1 mbar) (Fig. 2b and c, Supplementary Fig. 12). For the quasi in situ alkaline HER mechanism investigation (Fig. 4), the samples of W-ACs coated Ni foam were initially treated by the increasing negative potentials in glove box, and then the spent samples were transferred to the analysis chamber of NAP-XPS system for the detection of W 4f, O 1s, and valence band signals. Obviously, our work of alkaline HER mechanism investigation conducted in the vacuum-connected SPEC NAP-XPS system can avoid the air interference (O_2 adsorption) and any contamination, which was beneficial for detecting the intrinsic XPS signals induced by electrocatalysis. The photon source is the monochromatic X-ray source of Al $K\alpha$ (1486.6 eV); the overall spectra resolution is Ag 3d5/2, < 0.5 eV FWHM at 20 kcps@UHV; the pressure was 0.1 mbar (H_2) and UHV (5×10^{-9} mbar) conditions of the in situ variable temperature NAP-XPS measurements and quasi in situ alkaline HER mechanism investigation, respectively. The samples were in metallic states, because they have electronic bands across the Fermi level in valence band (Fig. 2c) (*Nat. Commun.* **11**, 4789 (2020)), meanwhile, the adsorption edge positions of all tungsten species (W-SAs, W-ACs, WC NPs) were located between W foil (0) and WO_2 (+4) with metallic character (Supplementary Fig. 14) (*J. Am. Chem. Soc.* **137**, 6983-6986 (2015)). All above-mentioned details have been added in the revised manuscript.

Comments 5:

Curve-fitting of the XPS data: Could the authors please specify what kind of lineshapes they used and which values they found for parameters such as peak position, gaussian width, lorentzian width etc., in all cases. It would be suitable if the parameters were provided in the supplementary information. I note that some lines look overly lorentzian in shape. Even background shapes and parameters should be stated.

Author reply: Thanks for the referee's valuable comments. According to the suggestion of the referee, we have refitted all XPS raw data requiring detailed analysis, and the corresponding parameters (peak position, FWHM, peak intensity) have been presented in the supplementary information. In particular, the XPS spectra was analyzed and deconvoluted using the Gaussian-Lorentzian sum function with a 30 % Gaussian-Lorentzian value to optimize the spectra, and a Shirley background correction was carried out to subtract the background noise.

Comments 6:

The calibration of the photoelectron spectra were done "to the standard C 1s peak (284.6 eV)", i.e. to what by many authors is referred the "carbonaceous carbon peak". The problem is - there is nothing such as "carbonaceous carbon". What is adsorbed on a sample as "sample contamination" depends on the sample material, its history and the sample environment from which residual gases are released (and then adsorbed on the sample). There carbonaceous carbon peak is by no means a reliable reference. As I written above, I guess the samples were sufficiently conductive to allow reliance on the calibration of the spectrometer alone. For conductive samples the calibration of the spectrometer should be done thoroughly and in periodic intervals, but then one can rely on it.

Author reply: Thanks for the referee's valuable comments. The spectrometer was regularly calibrated by Au, Ag, Cu standard samples (Calibration of zero energy point of Fermi level and linear calibration), and the tungsten samples were sufficiently conductive to allow reliance on the calibration of the spectrometer alone. In fact, we found that there was no need for calibration of the XPS data even comparing to the dominant peak of C 1s, because they were mainly centered at about ~284.6 eV, and the presented XPS profiles were indeed the raw data collected from the spectrometer. To avoid unnecessary misunderstanding, we have corrected the relevant description, which is described as follows:

"All binding energies were the raw XPS data without further calibration because the samples were sufficiently conductive to allow reliance on the calibration of the spectrometer alone"

Comments 7:

It is written that the Ni foam used in the XPS experiment was coated with carbon-supported tungsten single atoms. Could the authors please provide more details on how they did this?

Author reply: Thanks for the referee's valuable comments. Supplementary Fig. 1 illustrates the synthesis of W-SA powders and Ni foam coated with W-SAs. In brief, 3 g of polydopamine (PDA), 3g of P₁₂₃, 0.5 g of Tris, 0.6 g of Na₂H₂PO₄·2H₂O, and 0.04 g of Na₂WO₄·2H₂O were dissolved in distilled water (30 mL), then one piece of Ni foam was immersed into the hybrid solutions with continuous stirring for 6 h. As a result, the powder containing tungsten precursors and Ni foams coated with tungsten precursors were synthesized. After pyrolysis at 700 °C under Ar atmosphere and followed by alkaline leaching in 6 M KOH solution for 3 days, the desired W-SA powders and Ni foam coated with W-SAs were synthesized. Subsequently, W-AC@Ni foam and WC NPs@Ni foam were prepared at 550 and 650 °C, respectively. HAADF-STEM images confirmed that the tungsten species extracted from Ni foam by sonication are identical to those from those of the powder samples (Supplementary Fig. 7).

Supplementary Figure 1. (a) Schematic illustration of the preparation of tungsten-based powders and Ni coated with carbon-supported tungsten species. Typical HAADF-STEM images of (b)

W-SA powder, (c) W-AC powder, (d) WC NP powder. (e, f, g) The corresponding STEM images of tungsten species extracted from Ni foam by sonication.

Comments 8:

Supplementary Figure 6 and corresponding text in the manuscript: From the data shown here the intensities of the C 1s peak cannot be properly compared, since no proper background treatment/removal has been done. For the intensities to be comparable (at a glance) the backgrounds on the low- and high-binding energy sides must be the same for all spectra. In the case of the present spectra it would be advantageous to remove a polynomial or a Shirley background. With this said, a closer inspection indeed shows that the C 1s intensity decreases upon heating to 580 °C and increases again upon heating to 600 °C. With a proper data treatment this can be made visible in a much better way. With respect to the possible feature labelled "Carbide C": the signal-to-noise ratio does not really allow to make a definitive statement on its presence or absence. It might be there - or it might not. If the authors really want to prove its (expected) existence, they will have to take a spectrum with significantly better statistics.

Author reply: Thanks for the referee's valuable comments. According to the referee's suggestion, the C 1s core level XPS spectrum of tungsten species with better signal to noise ratio were collected at 500, 550, 580 and 600 °C, respectively. As shown in Supplementary Fig. 12, the C 1s XPS spectra exhibit an abruptly broad peak of the carbide C1s signal (282.7 eV) at 600 °C, which is extremely similar to the phenomenon reported by Zheng et al (*ACS Nano* **9**, 5125-5134 (2015)), indicating the increased amount of large-sized WC NPs existed in the sample. Meanwhile, we also observed that the C 1s intensity decreased upon heating to 580 °C and increased again upon heating to 600 °C, which was probably caused by the transition of graphitic carbon and carbide carbon in the aggregation of tungsten species. The relevant discussion has been added in the revised manuscript.

Supplementary Figure 12. C 1s core level XPS spectra of tungsten species collected at 500, 550, 580 and 600 °C, respectively.

Comments 9:

The authors curve-fitted the XPS data in Fig. 2b and, based on the curve-fitting results, they state that there is a shift to lower binding energy upon heating at 500 °C. The curve-fitting results are, however, not particularly good and a shift cannot really be inferred from them. A more reliable method would be to calculate the first moment of the W-C intensities (of either the 5/2 or the 7/2 peak or both, after background removal; a bit problematic is going to be the intensity of the W-O peak and of an unidentified structure on the high-binding energy sides of the W-C peaks). What is the unidentified structure? Could the authors please label the different lines?

Author reply: Thanks for the referee's valuable comments. According to the suggestion of the referee, we have refitted the W 4f XPS data, and the unidentified structure was also labeled in Fig. 2b. As can be seen, the initial state of the W-SAs exhibits three pairs of peaks after deconvolution, in which the major peaks with lower binding energies at 31.5 and 33.6 eV can be attributed to the intrinsic W-C bonds, while the second pair of peaks at higher binding energies (35.8 eV for W 4f_{7/2} and 37.9 eV for W 4f_{5/2}) are attributed to the W-O bonds (Supplementary Table 1). In addition to the two predominant pairs of peaks, another doublet with intermediate binding energies (32.6 and 34.8 eV) is also visible, which can be attributed to the carbon/oxygen-coordinated tungsten species due to the trace amount of oxygen in W-C coordination structure (W-C(O)) (*J. Mater. Chem. A*, **6**, 15395-15403 (2018); *ACS Appl. Mater. Interfaces* **12**, 22741-22750 (2020)). An obvious negative shift of the W 4f_{7/2} and W 4f_{5/2} in W-O species towards lower binding energies was found at 400 °C (Supplementary Table 1), indicating the bonded oxygen atoms were removed by hydrogen gas as the temperature increased. The negligible changes in the intrinsic W-C signals

were observed for the samples treated at 400 and 450 °C, indicating the W atoms might be stable in the single-atom state on the carbon substrates (*Phys. Rev. Lett.* **105**, 196102 (2010)). When the temperature reached upon 500 °C, the intensities of W-C(O) and W-O signals decreased significantly (Supplementary Table 2), implying that they were transformed into carbon-coordinated tungsten species with lower valence state, as determined by the abruptly enhanced intensities of the low-valence W-C signals at 500 °C (Supplementary Table 2). Encouragingly, the intermediate W-C(O) species were almost absent, and the highest intensity of W-C signals with sharp peaks was detected in the sample treated at 550 °C (Supplementary Tables 2 and 3), indicating the W-ACs may be formed because of the thermal migration of tungsten atoms at high temperature (*Phys. Rev. Lett.* **105**, 196102 (2010); *Matter* **3**, 509-521 (2020)). However, the W-C signals became weaker when treated at temperatures higher than 550 °C (also confirmed by the controlled experiments in Supplementary Figs. 10 and 11), indicating the surface of the generated W-ACs might have been simultaneously coated by a graphitic carbon layer (*ACS Catal.* **2**, 765-769 (2012)), which can be transformed into the carbide carbon in the crystallized phase at high temperatures. The generation of carbide carbon was evidenced by the C 1s XPS spectrum of the tungsten species treated at 600 °C, showing an obviously broad carbide C1s signal located at approximately 282.7 eV (Supplementary Fig. 12). The relevant discussion has been added in the revised manuscript.

Fig. 2 (b) W 4f core level XPS spectra for tungsten species treated in the temperature range of 400 - 600 °C.

Comments 10:

Supplementary Figure 12: Quite generally the background treatment is not particularly good in any of the panels (the background is too low on the high-binding energy side). The background

used in panel c is simply faulty, since it is far too low also on the low-binding energy side. It is essentially only due the poor background that a "P-W" component can be fitted. In order to really prove the existence of a P-W component, the authors will have to measure a spectrum with better statistics.

Author reply: Thanks for the referee's valuable comments. Yes, it is very challenging to detect the P-W signal, because the total P element was as low as ~2.6 wt% (ICP-OES) in the tungsten species, and most of the P atoms exists in the carbon substrate (P-C/O) rather than a P-W component, but efforts have been made in the revised manuscript. The P 2p core level spectrum of W-SAs, W-ACs and WC NPs were measured for comparison. As shown in Supplementary Fig. 18, the P 2p core-level XPS spectra of W-SAs and W-ACs only exhibited P-C (132.6 eV) and P-O (133.8 eV) bonds at high binding energies after deconvolution (*Angew. Chem. Int. Ed.* **54**, 6325-6329 (125)), whereas an additional P-W (129.1 eV) signal at relatively low binding energy was found for the WC NPs (*J. Mater. Chem. A*, **4**, 15327-15332 (2016)), indicating that tungsten atoms in W-SAs and W-ACs were not directly coordinated with P atoms in the carbon matrix. It should be noted that the XPS spectra were analyzed and deconvoluted using the Gaussian-Lorentzian sum function with a 30 % Gaussian-Lorentzian value to optimize the spectra, and a Shirley background correction was carried out to subtract the background noise.

Supplementary Figure 18. P 2p core level XPS spectra of (a) W-SAs, (b) W-ACs and (c) WC NPs.

Comments 11:

As is clear from my comments 8 and 10, I do not think that the XPS data can be used to prove the existence of a carbide for the WC-NPs or bond between P and W in the WC-NPs. This is not

necessarily a major problem, given that the material amount is small. But, of course, XPS data with a sufficient signal-to-noise ratio where the corresponding features can be identified unambiguously would strengthen the case of the authors. Hence, the authors might want to consider measuring such XPS data.

Author reply: Thanks for the referee's valuable comments. Yes, it is very challenging to detect the W-C carbide and W-P signals in WC NPs because of the relatively small amount of the target materials. However, XPS data with a sufficient signal to noise ratio has been collected in accordance with the referee's suggestion, and the corresponding results have also been discussed in comment 8 and 10.

Comments 12:

It is difficult to link the colours of the spectra in Fig. 2c to those in the legend, partly because different line thicknesses have been used. The authors could think of a better way of identifying the different spectra. Similarly, the colour scale in Supplementary Figure 7 is chosen in such a way that it is difficult to distinguish the spectra of C@Ni foam and of W-SAs at 600 °C.

Author reply: Thanks for the referee's valuable comments. In order to address the referee's concerns on the better identification of the different spectra, the relevant valence band have been treated with more concise and clear curves.

Comments 13:

It is standard convention to display XPS data with a reverse binding energy scale, i.e. with high binding energies to the left and low to the right. The authors might want to consider following this convention in order to make it easier for the readers to consider the data.

Author reply: Thanks for the referee's valuable comments. According to the referee's suggestion, all XPS data have been revised in binding energy scale with high binding energies to the left and low to the right.

Comments 14:

Fig. 2d and Fig. 4d: The simultaneous use of green and red colours without further means of identification than a colour legend is not advisable, since around 5 % of the world population (or around 10 % of the male population) have a red/green colour vision deficiency.

Author reply: Thanks for the referee's valuable comments. We have corrected the red/green colour profiles to satisfy the reading requirements, as shown in Figs. 2d and 4d.

Comments 15:

Could the authors please render the following sentence on page 2 clearer: "Usually, unsaturated coordination occurs in tungsten carbides having diameters of less than 5 nm (or single tungsten atoms) owing to the defects by their ultrasmall size."

Author reply: Thanks for the referee's valuable comments. We have revised the relevant sentence, which is described as follow:

"Usually, unsaturated coordination occurs in tungsten carbide nanoparticles (diameters < 5 nm) or single tungsten atoms because of the defective nature by their ultrasmall size."

Comments 16:

In Fig. 1 c a dashed red arrow is shown. In the text it is referred to as "the dashed orange arrow". Please correct.

Author reply: Thanks for the referee's valuable comments. The mistake has been corrected.

Comments 17:

Fig. 1f: The red arrows confused me a while until I understood that each of them indicates a structure that is identified with a W-AC. The caption text could be rendered clearer, e.g. "Enlarged STEM of a surface with W-ACs, which are indicated by red arrows." I am not quite sure that I understand how the W-ACs were identified. I can see quite much more structure in the image that looks the same to me. Could the authors please explain more thoroughly?

Author reply: Thanks for the referee's valuable comments. Multiple characterizations have been performed to comprehensively determine the structure of the as-obtained W-ACs. First, the magnified high-angle annular dark-field scanning transmission electron microscopy (HAADF-STEM) images observed by spherical aberration electron microscope clearly evidenced that a large proportion of the bright spots were adjacent to each other and presented the W-W structures, which were indicated by the red dashed circles (Fig. 1f and Supplementary Fig. 6). Second, As indicated by the valence band spectrum, the markedly enhanced intensity of the valence band near the Fermi level on W-ACs evidenced that W-W bonds began to appear when the temperature increased upon 500 °C (Fig. 2c) (*Nat. Mater.*, **17**, 1033-1039 (2018)). Finally, X-ray

absorption near-edge structure (XANES) and Fourier-transformed extended X-ray absorption fine structure (FT-EXAFS) are powerful techniques to explore the local structure of tungsten atoms (*Adv. Mater.* **30**, 1800396 (2018)). As a decrease in the oxidation state of tungsten species usually leads to a shift of the adsorption edge position to more negative energy in XANES spectra (*Nano Energy* **60**, 394-403 (2019); *Electrochim. Acta*, **283**, 834-841 (2018); *Korean J. of Chem. Eng.*, **8**, 164-167 (1991)), a slightly negative shifting in the adsorption edge for W-ACs with respect to W-SAs indicates a relatively high reduction in W-ACs due to the formation of W-W bonds (Supplementary 14). The best fitting results of the R space in Fourier-transformed extended XAFS (FT-EXAFS) quantitatively demonstrate that the W-ACs exhibited a W-W coordination number of 1.1, which is higher than that of atomically dispersed W-SAs, but lower than WC NPs. The W-W coordination number very close to 1 implies that most of W atoms are coordinated with one W atom over the entire P-doped carbon substrates (Supplementary Fig. 16, Supplementary Table 4). The relevant discussion has been added in the revised manuscript.

Supplementary Figure 6. (a, b, c) Typical STEM images of W-ACs, where W-W structures are observed and highlighted by red dashed circles; (d) Valence band collected by in situ variable temperature NAP-XPS measurements; (e) W L_{3} -edge XANES and (f) FT-EXAFS in R-space for W-ACs, where the solid line is the fitting curve.

Comments 18:

Fig. 1f: The inset is a bit funny in that it represents more or less the same magnification as the main image (the 5 Å length bar is around half the size of the 1 nm = 10 Å length bar). What is the

meaning of the inset then? What is it that I am supposed to see and what extra information am I supposed to gain?

Author reply: Thanks for the referee's valuable comments. The inset in Fig. 1f was presented aiming at a clear observation of the neighboring W atoms with closer distance in comparison with W-SAs. We have added a much clearer image of W-ACs in Fig. 1f and Supplementary Fig. 6.

Fig. (a, b, c) Typical STEM images of W-ACs, where the W-W structures are observed and highlighted by red dashed circles.

Comments 19:

Fig. 1a, Supplementary Figures 9, 13, 19: Could the authors please identify the colours used for the different elements? (this has been partly done in some of the figures, but not in all, and where it has been done it is not easy to see)

Author reply: Thanks for the referee's valuable comments. We have identified the colors used for the different elements in Supplementary Figs. 15 and 28.

Comments 20:

The scales of Figs. 1b, e and h are very different scales, which makes it difficult to compare the morphologies. It would be better if images with identical sizes were compared. The comment also extends to the images in Supplementary Figures 3 and 4.

Author reply: Thanks for the referee's valuable comments. According to the referee's suggestion, Figs. 1b, e and h have been corrected with the same scale bar in the revised manuscript (Figs. 1 b, e, h).

Fig. 1 Typical HAADF-STEM images of (b) W-SAs, (e) W clusters, and (h) WC NPs.

Supplementary Fig. 4a presented at large scale bar (50 nm) was to illustrate the overall morphology of the nanosheet substrates. Only carbon layers without any nanoparticles can be seen in the HRTEM (scale bar: 10 nm) image of W-SAs, probably because it is challenging to observe ultrasmall nanocrystals (i.e., diameters less than 1 nm) using TEM (Supplementary Fig. 4b). HAADF-STEM (scale bar: 2 nm) image of W-SAs confirmed the existence of the single tungsten atoms (Supplementary Fig. 4c). The above presentation strategy of single atoms using low- and high-magnification TEM/STEM images has been widely validated by previous reports (*Angew. Chem. Int. Ed.* **55**, 1-6 (2016); *ACS Nano* **11**, 6930-6941 (2017)). In addition, Supplementary Fig. 5b and Fig. 1h have been rearranged with exchanged positions using the same scale bar (Supplementary Fig. 5).

Supplementary Figure 4. (a) Low- and (b) high-magnification transmission electron microscopy (TEM) images of W-SAs. (c) Atomic-level high-angle annular dark-field scanning transmission electron microscopy (HAADF-STEM) image of W-SAs.

Supplementary Figure 5. HAADF-STEM images of (a) W-ACs and (b) WC NPs.

Comments 21:

Supplementary Figures 9b and 13a are identical - therefore, I would like to recommend combining the two Supplementary Figures 9 and 13 into a single one.

Author reply: Thanks for the referee's valuable comments. According to the referee's suggestion, we have combined the two supplementary Figs. 9 and 13 into a single one (Supplementary Fig. 15).

Supplementary Figure 15. Changes in Bader charges of W atom in (a) W-SAs, (b) WC NPs, (c) P-doped W-ACs, and (d) W-ACs without P doping.

Comments 22:

Supplementary Figure 13: Even in a Bader charge analysis there is an uncertainty, since the derived Bader charges will e.g. depend on which functional was used in the DFT calculation. The differences in Bader charges on the tungsten atoms in the presence and absence of P in Supplementary Figure 13 are small. I would like to ask the authors to provide a qualified (and referenced) statement in the manuscript to which extent the difference in Bader charges is significant.

Author reply: Thanks for the referee's valuable comments. Yes, we agree with the referee's viewpoint that the calculated values of Bader charges usually depend on the functional used. All Bader charges presented in the text were calculated by the generalized gradient approach (GGA) of the Perdew-Burke-Ernzerhof (PBE) functional (*Phys. Rev. Lett.* **77**, 3865-3868 (1996)), in order

to check the influence of different function on the charge of the W atoms, we then recalculated the Bader charges using another generalized functional of local density approximation (LDA) (*Phys. Rev. Lett.* **102**, 026101 (2009)), and the corresponding values extracted from W-ACs with and without P doping were presented in Supplementary Fig. 15e and f. It is clear that the W atom near P atom still suffers larger amount of depleted electrons than the others. Therefore, based on above calculation, we can conclude that the introduction of P atom as a dopant indeed modifies the electronic structures of the W-W structures.

In addition, the small change of above calculated Bader charges was found to be significant for HER activity as presented by previous reports. For example, Zhu et al. reported the WS₂/MoS₂ heterostructure for electrocatalytic HER process (*J. Appl. Phys.* **120**, 024301 (2016)), in which Bader charge analysis revealed that a charge transfer of 0.014 e from WS₂ to MoS₂ will inevitably facilitate the overall HER process. Moreover, Huang et al. reported a volcano relationship between the hydrogen adsorption ability (HER activity) and Bader charges carried by Ru11, which demonstrated that the the small change (0.0172 e) of Bader charges in Zn-doped Ru11 can make the HER activity close to the volcano peak in comparison with that of Fe doping (*Appl. Surf. Sci.* **556**, 149801 (2021)). Therefore, our small change (>0.04 e) of Bader charges can also be used to predict the improved alkaline HER activity on P-doped W-ACs.

Supplementary Figure 15. Change in Bader charges of (a, c) P doped W-ACs and (b, d) W-ACs without P doping, in which the values of Bader charge in (a and b) were calculated by the PBE functional, whereas the LDA functional was used to calculate the ones in (c and d).

Comments 23:

Supplementary Figure 18: The W 4f presented here for W-SAs before and after HER are helpful for assessing to which extent they are oxidised. It would be helpful if the same kind of data could be shown for the W-ACs and even the WC-NPs.

Author reply: Thanks for the referee's valuable comments. According to the referee's suggestion, the W 4f core level XPS spectra of W-ACs and WC NPs before and after alkaline HER process have also been examined (Supplementary Fig. 26). As can be seen, both W-SAs and WC NPs exhibited strong W-O peaks after alkaline HER process. In particular, one could see the markedly decreased W-C signals in W-SAs, indicating the W-SAs have suffered from serious oxidation problems after alkaline HER process, whereas the intrinsic W-C signals without obviously enhanced W-O peaks were detected on W-ACs, highlighting the improved structural robustness of W-ACs in alkaline electrolyte with respect to the W-SA counterparts.

Supplementary Figure 26. W 4f core level spectra of (a) W-SAs, (b) W-ACs, and (c) WC NPs before and after alkaline HER process.

Comments 24:

On page 9, the authors state that "the white-line intensities [in the XANES] spectra for all tungsten species are high than that of the W foil reference, suggesting that all tungsten species are in the oxidation state." Which oxidation state? (even 0, for a metallic atom, is an oxidation state!)

Author reply: Thanks for the referee's valuable comments. It should be noted that single tungsten atom (W-SAs) and tungsten atomic clusters (W-ACs) are anchored on the P-doped carbon atoms

via carbon coordination, thus the W-C chemical bonds have been formed in W-SAs and W-ACs, which implies the existence of electron gain and loss between W and C atoms, as evidenced by the analysis of Bader charge (Supplementary Fig. 15). Moreover, a decrease in the oxidation state of tungsten species usually leads to a shift of the adsorption edge position to more negative energy in XANES spectra based on previous reports (*Nano Energy* **60**, 394-403 (2019); *Electrochim. Acta*, **283**, 834-841 (2018); *Korean J. of Chem. Eng.*, **8**, 164-167 (1991)). Accordingly, W foil (0) and WO₂ with metallic character (+4, synthetic details seen Supplementary information) as the references were used to determine the oxidation states of the as-obtained W-SAs, W-ACs, and WC NPs materials (Supplementary Fig. 14). As can be seen, the adsorption edge positions of the as-obtained tungsten species are between W foil (0) and WO₂ (+4), suggesting the average oxidation states of W-SAs, W-ACs, and WC NPs are located between 0 and +4 following the order of WC<W-ACs<W-SAs. The relevant discussion has been added in the revised manuscript.

Supplementary Figure 15. Change in Bader charges of W atom in (a) W-SAs, (b) W-ACs and (c) WC NPs. (d) The enlarged adsorption edge position in the XANES spectra of W foil, WO₂, WO₃, W-SAs, W-ACs, and WC NPs.

Comments 25:

By convention, XANES spectroscopy in the soft x-ray regime is called either NEXAFS spectroscopy or x-ray absorption spectroscopy (XAS). The author might want to consider to

adhere to the convention when it comes to the C K-edge NEXAFS spectra, which they call XANES spectra.

Author reply: Thanks for the referee's valuable comments. According to the referee's suggestion, we have renamed the title of X-ray adsorption spectroscopy as C K-edge NEXAFS spectra in Supplementary Fig. 12.

Comments 26:

On page 10, referral is made to a "Table 1". This should be "Supplementary Table 1".

Author reply: Thanks for the referee's valuable comments. We have corrected the mistake.

Comments 27:

It is found that the doping with P plays a very significant role for the HER activity of the W-ACs (cf. Supplementary Figure 17). Hence, a proper comparison would be to P-doped Pt/C rather than standard commercial Pt/C. In the same vein, comparison should be made not to "bare C", but to P-doped "bare C" (Fig. 3a) (and the latter comparison is more important than the former to assess the activity of the W-ACs). I have not made a thorough literature search for such systems and their activity, but a quick search brought directly up a report on an increased activity of a P (and N) co-doped Pt/C material, albeit in acidic medium (Wang et al., *Nano Research* 10 (2017) 238).

Author reply: Thanks for the referee's valuable comments. Sorry, we have made a expression error of the bare C, which is indeed the P-doped carbon material (P-doped C), and we have modified the relevant sentence using P-doped C in the revised manuscript. In order to address the referee's concern on the comparison of alkaline HER activity between the W-ACs and commercial P-doped Pt/C, two types of P-doped Pt/C catalysts were synthesized: one was obtained by the low-temperature phosphidation of commercial Pt/C (denoted as P-doped Pt/C I) (*J. Mater. Chem. A*, **4**, 9691-9699 (2016)), and the other (denoted as P-doped Pt/C II) was collected according to the synthetic method of tungsten precursors with PtCl₂ reagent replacing Na₂WO₄·2H₂O (More details seen in the Supplementary Fig. 22). Subsequently, the LSV curves of P-doped Pt/C I and P-doped Pt/C II were measured in 1 M KOH electrolyte, in which P-doped Pt/C I exhibited more superior alkaline HER activity ($\eta_{10}=25$ mV) than those of P-doped Pt/C II ($\eta_{10}=30$ mV) and commercial Pt/C ($\eta_{10}=35$ mV). The relevant profiles (LSV and Tafel plot) of P-doped Pt/C I have been added in Fig. 3a and c for a clear comparison.

Supplementary Figure 22. TEM images of (a) P-doped Pt/C I and (b) P-doped Pt/C II, with the corresponding HRTEM images in the insets. (c) The EDX spectrum of P-doped Pt/C I and P-doped Pt/C II, where the C, O, Pt, and P are from the materials, whereas the Cu element originates from the copper grid. (d) LSV curves and (e) Tafel plots of W-ACs, fresh commercial Pt/C, P-doped Pt/C I, and P-doped Pt/C II.

Comments 28:

Fig. 3b lacks references. I guess the data stem from the works cited on line 10 of page 12?

Author reply: Thanks for the referee's valuable comments. We have added the citation of relevant references in the corresponding figure caption in the revised manuscript (Fig. 3b).

Fig. 3 (b) The overpotentials at 10 mA/cm² extracted from the W-ACs and P-doped Pt/C catalysts in this work, in comparison with those of non-noble metal catalysts previously reported in alkaline media^{3,6,10,16,23,32-44}.

Comments 29:

I guess "JCPDF" on page 5 should read "JCPDS" (which today actually is the ICDD).

Author reply: Thanks for the referee's valuable comments. The name of the standard ICDD card has been corrected to JCPDS in the revised manuscript.

Replies on comments of reviewer 2:

This work has developed a W atomic cluster (W-ACs) supported on P-doped carbon materials as the electrocatalysts of alkaline HER. The coordination environment and electronic structure of W-AC have been characterized. A high HER activity close to commercial Pt/C catalysts was obtained on W-ACs. DFT calculations were performed to rationalize the high activity of W-AC. However, there are some critical questions in theoretical calculations.

Comments 1:

As known, the structure of metal cluster is often not well defined. In DFT calculations, why do you select a double-atom model for W-ACs, namely two adjacent W atoms anchored on the graphene substrate? Typically, metal cluster can contain dozens of atoms, hence a double-atom model is not reasonable.

Author reply: Thanks for the referee's valuable comments. Yes, the definition of metal cluster is very challenging, but multiple characterizations have been performed to comprehensively determine the structure of the as-obtained W-ACs for the establishment of DFT model. First, the magnified high-angle annular dark-field scanning transmission electron microscopy (HAADF-STEM) images observed by spherical aberration electron microscope clearly evidenced that a large proportion of the bright spots are adjacent to each other and presented the W-W structures, as indicated by the red dashed circles (Fig. 1f and Supplementary Fig. 6). Second, as indicated by the valence band spectrum, a remarkable enhancement of the intensity of the valence band near the Fermi level on W-ACs evidenced that W-W bonds began to appear when the temperature increased upon 500 °C (Fig. 2c) (*Nat. Mater.*, **17**, 1033-1039 (2018)). Finally, X-ray absorption near-edge structure (XANES) and Fourier-transformed extended X-ray absorption fine structure (FT-EXAFS) are powerful techniques to explore the local structure of tungsten atoms (*Adv. Mater.* **30**, 1800396 (2018)). As a decrease in the oxidation state of tungsten species usually leads to a shift of the adsorption edge position to more negative energy in XANES spectra (*Nano*

Energy **60**, 394-403 (2019); *Electrochim. Acta*, **283**, 834-841 (2018); *Korean J. of Chem. Eng.*, **8**, 164-167 (1991)), a slightly negative shifting in the adsorption edge for W-ACs with respect to W-SAs indicates a relatively large degree of reduction state for tungsten species in W-ACs due to the formation of W-W bonds. Moreover, the best fitting results of the R space in Fourier-transformed extended XAFS (FT-EXAFS) quantitatively demonstrated that the W-ACs exhibited a W-W coordination number of 1.1, which is higher than that of atomically dispersed W-SAs, but lower than WC NPs. The W-W coordination number very close to 1 implies that most of W atoms are coordinated with one W atom over the entire P-doped carbon substrates (Supplementary Fig. 16, Supplementary Table 4). In fact, the comprehensive characterizations using spherical aberration electron microscope, XPS, and EXAFS techniques to identify the structure of the atomic-level materials have been developed as a general strategy in single atom/cluster catalysis (*Nat. Commun.* **8**, 1070 (2017); *Angew. Chem. Int. Ed.* **58**, 2321-2325 (2019); *Matter* **3**, 509-521 (2020)). Therefore, the double-atom model representing W-W structure for the DFT calculations should be reasonable on the basis of our experimental observations.

Fig. (a, b, c) Typical STEM images of W-ACs, where the W-W structures are observed and highlighted by red dashed circles; (d) Valence band collected by in situ variable temperature NAP-XPS measurements; (e) W L_3 -edge XANES and (f) FT-EXAFS in R-space for W-ACs, where the solid line is the fitting curve.

Comments 2:

In experiments, Pt/C exhibits a higher HER activity than W-ACs (Figure 3a-c), while W-ACs shows much lower water dissociation barrier than Pt (Fig. 5d), which are inconsistent. Although H* desorption on Pt is a little easier than on W-ACs, water dissociation is the rate-determining step on Pt, with a barrier of 0.81 eV.

Author reply: Thanks for the referee's valuable comments. We only considered the (111) facet of Pt metal as active sites for electrocatalysis in the theoretic simulations because of its optimal hydrogen adsorption strength (*J. Am. Chem. Soc.* **138**, 16174-16181 (2016)), but the commercial Pt/C powder-catalyzed alkaline HER process actually involved the synergistic work of multiple crystalline planes, such as (111), (100), (110), (311) and so forth, as observed by the transmission electron microscopy (TEM). As shown in the low-magnification TEM image, a large amount of Pt NPs with sizes less than 5 nm were uniformly dispersed on the carbon substrates (Supplementary Fig. 30a), and the corresponding selected area electron diffraction (SAED) pattern (inset in Supplementary Fig. 30a) showed the polycrystalline nature of the nanoparticles with the presence of (111), (200), (220), and (311) facets, respectively. The marked d-spacing values of 0.23, 0.20, and 0.14 nm in the HRTEM images are well in agreement with the (111), (200), and (220) planes of Pt NPs, respectively (Supplementary Fig. 30b-e). Accordingly, the energy barriers in water dissociation and hydrogen desorption steps on above observed surfaces were calculated for comparison. The resultant values of water dissociation barriers over (111), (100), (110), and (311) facets of Pt were 0.81, 0.83, 0.69, and 0.43 eV, respectively (Supplementary Fig. 30f), indicating the Pt (311) has the lowest energy barrier in water dissociation step and the energy barrier of Pt in the rate-determining step (water dissociation) has been substantially weakened, which also confirms that the Pt (111) is generally inefficient for the cleavage of H-OH bonds (*Angew. Chem. Int. Ed.* **58**, 13107-13112 (2019); *Nano Energy* **81**, 105636 (2021)). However, the calculation in the subsequent hydrogen desorption step evidenced that Pt (111) takes the fastest hydrogen desorption process with energy barrier as low as 0.28 eV, which is much lower than those of other crystalline planes (Supplementary Fig. 30g). In addition, another three advantages of commercial Pt/C powders that were neglected by DFT calculations should also be emphasized for boosting alkaline HER activity: (i) more available and highly active sites for Pt-catalyzed HER process could be introduced due to their ultrasmall size (< 5 nm) (*J. Am. Chem. Soc.* **139**, 5285-5288

(2017); *Nat. Commun.* **7**, 13216 (2016)); (ii) the observed twin boundary (TB) of Pt (111) facet can efficiently boost the electrocatalytic activity (Supplementary Fig. 30c), because they can provide more low-coordination atomic sites for alkaline HER process, which are beneficial for electrocatalysis (*J. Am. Chem. Soc.* **143**, 4387-4396 (2021)); (iii) the energy barriers of alkaline HER process can be regulated by the synergistic work of carbon layer and encapsulated Pt NPs (*Angew. Chem. Int. Ed.* **54**, 2100-2104 (2015); *Nano energy* **52**, 494-500 (2018)). Therefore, according to the calculated energy barriers in the two-step alkaline HER process and previous reports, we can conclude that multiple crystal planes, ultrasmall size, twin boundary, and carbon-Pt interface work in synergy to alter the energy barriers for the observed excellent alkaline HER activity on commercial Pt/C catalyst.

Supplementary Figure 30. (a) Low TEM image of Pt NPs on the carbon substrates with the corresponding SAED pattern in the inset. (b, c, d, e) HRTEM images of the lattice fringes in Pt, with the twin boundary indicated by the red arrow in (c). The free energy diagrams for (f) water dissociation and (g) hydrogen desorption steps on (111), (100), (110), and (311) facets of Pt.

Comments 3:

In DFT calculations, the adsorbed H₂O was considered as the initial state (IS) for water dissociation process. However, H₂O in solution can dissociate into adsorbed H* and OH⁻ directly, namely H₂O(l) → H* + OH⁻(l), which is more reasonable than the adsorbed H₂O dissociation (H₂O* → H*+OH*) in alkaline HER.

Author reply: Thanks for the referee's valuable comments. Yes, H₂O in acidic and neutral solutions can also dissociate into H⁺ (then adsorbed on active sites, *H) and OH⁻ directly. However, unlike the acidic and neutral conditions, the split H⁺ in alkaline electrolyte will quickly react with the OH⁻ ions for the generation of H₂O, which will be very hard to adsorb on the catalyst surface. Moreover, according to the reaction mechanism of alkaline HER process, there will be one mole electrons to be consumed for forming one mole *H intermediates in water dissociation step (*Nat. Commun.* **5**, 4695 (2014); *Nat. Commun.* **8**, 15437 (2017)), obviously, no electrons will be consumed without the adsorption of H₂O molecules on the catalyst surface in the initial state (Volmer step). In particular, the OH⁻-catalyst interaction has been considered as an inevitable factor for the alkaline HER process (*Nat. Mater.* **11**, 550-557 (2012); *Nat. Commun.* **8**, 15437 (2017); *Energy Environ. Sci.*, **9**, 2789-2793 (2016); *ACS Catal.* **10**, 7322-7327 (2020)), which can be tracked by our quasi in situ alkaline HER mechanism investigation using W 4f XPS signals (the change of W-O signals caused by OH^{*} adsorption) and valence band (intensity decrease and recovery), revealing the change of the adsorption behavior of the adsorbed OH^{*} intermediates on W-ACs after treatment by increasing negative potentials (Fig. 4a-d, Supplementary Tables 5-8). Meanwhile, W-SAs have suffered from serious oxidation problems after alkaline HER process (Supplementary Fig. 26a), which was probably attributed to the adsorbed OH^{*} rather than the OH⁻ ions in electrolyte, because the Coulomb effect of cathode electrode can prevent OH⁻ (electrolyte)-induced corrosion. Therefore, considering the alkaline HER reaction mechanism (electrolyte and electron consumption), the OH^{*} adsorption of previous reports, and our experimental observations, the established models of adsorption for the H₂O and OH^{*} on the W-ACs should be reasonable.

Fig. 4 (a) W 4f core level XPS spectra of W-ACs treated at 0.00, -0.02, -0.04, and -0.06 V versus

RHE and soaked in KOH solution, respectively. (b) The corresponding valence band. (c) W 4f core level XPS spectra of W-SAs before and after alkaline HER process.

Comments 4:

Why P-doped W-ACs shows better activity than the W-ACs without P-doping? What the role of P-doping? Although the authors have analyzed Bader charge (Fig. S13), the free energy diagram of HER on W-ACs without P-doping is also needed to compare with the P-doped sample.

Author reply: Thanks for the referee's valuable comments. The P-doped W-ACs indeed exhibited more superior alkaline HER activity than that of the sample without P-doping in 1 M KOH electrolyte (Supplementary Fig. 25a and b), which can be attributed to the following two reasons: (i) the P-doping can accelerate the charge transfer during the alkaline HER process, due to the highly polarized P-C bond formed between P and C with different electronegativities, which will serve as an electron transfer bridge to improve the overall conductivity (*Nat. Commun.* **9**, 2533 (2018); *Adv. Sci.* **8**, 2101314 (2021)). Meanwhile, the excellent conductivity of W-ACs with P-doping was also evidenced by the smaller charge transfer resistance in comparison with that of the sample without P-doping (Supplementary Fig. 25c), which could boost the prior water dissociation step and the accumulation of the electrons in cathode, thus accelerating the desorption of OH* intermediates (supplementary Fig. 26d), (ii) the P-doping can regulate the electronic structure of the W-W structures for appealing alkaline HER process, as evidenced by the Bader charge and free energy diagram analysis (Supplementary Figs. 15 and 31). As shown in Supplementary Fig. 31, the energy barrier of the prior water dissociation on the W-ACs sample without P doping (0.28 eV) is higher than that of P doped W-ACs (0.24 eV), and the positive value of H* adsorption energy suggested the weak binding energy of H* intermediates on the catalyst surface of W-ACs without P doping.

Supplementary Figure 25. (a) Nyquist plots of W-ACs with and without P-doping. (b) W 4f core level XPS spectra of W-ACs with and without P-doping before and after HER process. (c) The corresponding calculated free energy diagrams for (d) water dissociation and (e) hydrogen desorption steps on W-ACs with and without P-doping.

Comments 5:

For molecule H₂ formation from adsorbed H*, the energy barrier (activation energy) should be calculated to evaluate the reaction kinetics.

Author reply: Thanks for the referee's valuable comment. According to the referee's suggestion, the energy barriers of H₂ formation from adsorbed H* (activation energy) on W-ACs and Pt (111) (the optimal facet of hydrogen desorption) have been calculated. As shown in Supplementary Fig. 32a, a pair of W-W structures were constructed to calculate the energy barrier of molecule H₂ formation, in which the molecule H₂ was generated between the W-W structures rather than the W atoms, because the W sites have strong dissociation ability of molecule H₂ (*Nat. Commun.* **11**, 4789 (2020)). As a result, the energy barrier was calculated to be approximately 1.43 eV (Supplementary Fig. 31b), which is higher than that of Pt (111) (0.88 eV) (*Catalysts*, **8**, 450 (2018)). Therefore, combining previous small energy barriers in water dissociation (0.24 eV) and H* desorption (-0.31 eV) on W-ACs, we can conclude that the relatively high energy barrier in molecule H₂ formation should be further regulated on W-ACs in our future work. Meanwhile, as raised in comments 2 by the referee, the relatively large energy barrier of molecule H₂ formation should be considered as a significant reason to explain why the alkaline HER activity of W-ACs is relatively inferior to commercial Pt/C in 1 M KOH electrolyte.

Besides, in order to directly display the high activity of hydrogen production using the synthesized W-ACs catalyst, a video recording the formation, growth and detachment of electrogenerated hydrogen gas bubbles at the surface of W-ACs at low potentials has been provided in the revised manuscript (Supplementary Fig. 32c). The gas production was collected for further gas chromatographic analysis, evidencing that the component of the as-obtained gas was H₂ (Supplementary Fig. 32d).

Supplementary Figure 32. (a) The H* intermediates (IS), transition states (TS), and products of H₂ desorption on W-ACs (FS). (b) The calculated free energy diagram for molecule H₂ formation from adsorbed H* based on the calculated models of W-W structures and Pt (111). (c) Optical images showing the generation and detachment of H₂ bubbles under the increasing overpotentials (η). (d) The component analysis of gas bubbles detached from W-AC catalyst by gas chromatography.

Comments 6:

How did you calculate the adsorption free energy of OH*? Was the adsorption free energy referenced to hydroxyl (OH⁻) in solution, or neutral OH in vacuum (namely DFT calculated OH energy), or water? The OH* adsorption energy on W-ACs reached to -1.42 eV, indicating the desorption of OH* is very difficult on ambient condition, blocking the HER.

Author reply: Thanks for the referee's valuable comments. The adsorption free energy of OH* on the catalyst surface (W-SAs, W-ACs, WC NPs) is calculated by the reference energies of H₂O and H₂ in the gas phase (*Energy Environ. Sci.*, **9**, 2789-2793 (2016)), and the calculated adsorption free energies of OH* were used to predict the change of interaction between OH* and the surface

of the as-obtained three types of tungsten species. The calculated adsorption free energy of OH* on W-ACs is -1.42 eV, which is lower than that of the W-SAs (-1.65 eV) (Supplementary Table 9), evidencing the formation of W-W bonds can weaken the adsorption free energy of OH* intermediates under ambient conditions (open circuit potential, OCP). Undoubtedly, the OH* adsorption on W-ACs is still strong, and such a strong adsorption affinity will block the HER active sites under alkaline electrolyte. However, the alkaline HER process driven by the increasing negative potentials was neglected by DFT calculations, where the detachment of OH* intermediates would be facilitated by the continuously enriched negative charge at cathode, as evidenced by the quasi in situ XPS characterization. As shown in Fig. 4b, the initial state of the sample only soaked in KOH electrolyte exhibited a relatively weak W-O signals corresponding to the adsorbed OH⁻ and robust lattice oxygen species, whereas a remarkable enhancement of the W-O signals was detected on the sample treated at 0 V (vs, RHE) (Fig. 4b, Supplementary Table 6), indicating the adsorption of the OH* intermediates on the catalyst surface. As expected, the intensities of the W-O signals reduced with increasing negative potential from -0.02 to -0.06 V, suggesting the OH* intermediates can be easily detached from the W sites at rather low potentials (Supplementary Table 6). In addition, the adsorption strength of OH* on W-SAs, W-ACs, and WC NPs were also examined by the oxidative linear sweep voltammetry (LSV) in 1 M KOH electrolyte (Supplementary Fig. 33). As can be seen, the potential for surface OH⁻ adsorption on W-SAs is more negative than that on W-ACs and WC NPs, implying the relatively strong adsorption of OH* on W-SAs (*Angew. Chem. Int. Ed.* **57**, 1944-1948 (2018), since the formation of W-W bonds can efficiently facilitate the desorption of OH* intermediates in the alkaline HER process.

Supplementary Figure 33. (a) W 4f core level XPS spectra of W-ACs treated at 0.00, -0.02, -0.04,

and -0.06 V versus RHE with the soaked sample in KOH solution (Initial state) as the reference, respectively. (b) oxidative linear sweep voltammetry (LSV) of W-SAs, W-ACs, and WC NPs in 1 M KOH electrolyte.

Replies on comments of reviewer 3:

The manuscript reported a tungsten atomic cluster (W-AC) catalyst for efficient alkaline HER. The authors apply a novel strategy to control the scale of tungsten atoms by thermal migration and prove that tungsten atoms could migrate and be aggregated into clusters at specific temperature (above 550 °C) via in-situ NAP-XPS. This work is quite impressive and it seems that these operando characterization experiments were carefully designed despite the difficulty of X-ray based operando analysis. Along with several experimental clues, DFT simulation also gives an adequate explanation for why 'W-ACs' facilitate alkaline HER catalysis. I have several questions and comments as follows.

Comments 1:

From the FT-EXAFS spectra at W L₃-edge (Fig. 2d), the W-C/O scattering path was assigned around 1.5~2 Å. However, W-C scattering path in the tungsten carbide is usually placed about 2.1 Å (Nat Commun 2016, 7, 13216, The author already cited). The coordination of W-substrate carbon and W-C coordination in tungsten carbide should be distinguished.

Author reply: Thanks for the referee's valuable comments. Yes, the W-C bonding length in tungsten carbide nanoparticles is usually placed about 2.1 Å (*Nat. Commun.* **7**, 13216 (2016)), whereas the presented W-C/O scattering path at around 1.5~2 Å was the raw data rather than the actual bonding length of W-C/O (Fig. 2d). Usually, the W-C/O bonding length in tungsten carbide can be obtained by fitting the broad peak of the raw data using the ARTEMIS module of IFEFFIT software packages, and the procedure will automatically conduct the phase correction of 0.2~0.5 Å on the basis of 1.5~2 Å (*Phys. Rev. B* **59**, 948-957 (1999)). In order to address the referee's concern on discrimination of the coordination of W-substrate carbon (W-SAs, W-ACs) and W-C coordination in tungsten carbide (WC NPs), we have fitted the R space of the three types of tungsten samples, which exhibit similar W-C bonding length (Supplementary Table 4) due to the similar carbide (W-C) features, as evidenced by W 4f core level XPS spectrum for the as-obtained W-SAs, W-ACs, and WC NPs (Fig. 2b). The observations imply that the substrate carbons bonded

to W-SAs and W-ACs indeed have the carbide nature of WC NPs. Similar phenomenon was also reported by Xie's work (*Matter* **3**, 509-521 (2020)), which demonstrated that FeN₄ single atoms and Fe₂N₆ clusters exhibited the same bonding length in comparison with that of Fe-N nanoparticles. Meanwhile, it should be noted that W-SAs have an obviously small bonding length (1.72 Å), but it was absent in W-ACs and WC NPs, indicating the formation of W-W bonds might stretch the W-C (substrate) bonding length towards the alternation of W-C carbide nature.

Comments 2:

From the W L₃-edge XANES (Figure S8), not only the pre-edge position, the white line intensity of WC-NP should be also considered to evaluate the valence states. The author mentioned that WC-NP is more reduced than W-SA and W-AC due to the lower edge shift, but white line intensity of WC-NP is larger than those of them, suggesting that the tungsten in WC-NP is more oxidized.

Author reply: Thanks for the referee's valuable comments. Sorry, there is a small mistake for the name of the pre-edge position, which should be modified as 'adsorption edge position' according to previously report (*Korean J. of Chem. Eng.*, **8**, 164-167 (1991)). Both the adsorption edge position and white line intensity in W L₃-edge XANES have been used to evaluate the valence states of tungsten element in tungsten-based compounds (*Nano Energy* **60**, 394-403 (2019); *Nanoscale*, **10**, 3469-3479 (2018); *Adv. Mater.* **30**, 1800396 (2018)). However, the white line intensity in tungsten species indicates the electronic density of the unoccupied d orbitals rather than directly representing the valence states (*Electrochim. Acta*, **283**, 834-841 (2018)), where the valence state is mainly determined by the occupied electrons in outer d and s orbitals. Usually, various factors should be considered when the white line intensity is changed, such as the valence state (*Adv. Mater.* **30**, 1800396 (2018)), the distortion of the tungsten-ligand local geometries (*J. Solid State Chem.* **178**, 1533-1538 (2005)), and the electron transition from W 2p_{3/2} orbitals to the unoccupied W 5d orbitals (*J. Alloys Compd.*, **857**, 157532 (2021)), so it should be combined with other evidences for the determination of valence states. An XANES study of tungsten carbides has demonstrated that the shift of the adsorption edge position can be used to better understand the change of oxidation state of tungsten in tungsten-based samples (W foil, WC, WO₃) (*Korean J. of Chem. Eng.*, **8**, 164-167 (1991)). Accordingly, W foil (0) and WO₂ with metallic character (+4) as

the references were used to determine the oxidation states of the as-obtained W-SAs, W-ACs, and WC NPs materials (Supplementary Fig. 14). As can be seen, the adsorption edge positions of the as-obtained tungsten species are between W foil (0) and WO_2 (+4), suggesting the average oxidation states of W-SAs, W-ACs, and WC NPs are located between 0 and +4 following the order of $\text{WC} < \text{W-ACs} < \text{W-SAs}$. Therefore, our conclusion on the valence states of W-SAs, W-ACs, and WC NPs based on the shift of adsorption edge position should be reasonable.

Supplementary Figure 14. (a) X-ray absorption near-edge structure (XANES) spectra of W-SAs, W-ACs, WC NPs, and the reference sample of W foil and WO_2 powder. (b) XRD pattern of the as-obtained WO_2 powder.

Comments 3:

In the constructed W-SA and W-AC models for DFT calculation, both models only contain one P atom which is replaced with C atom in a 6×6 supercell of graphene. A P atom in carbon substrate also might give an electronic effect on the binding energy of HER intermediate on the W active site, and the Bader charge of an adjacent tungsten atom can be calculated differently depending on the number of doped P atoms. Is there any specific reason for this composition?

Author reply: Thanks for the referee's valuable comments. We agree well with the referee that P atom in carbon substrate would give an electronic effect on the binding energy of HER intermediate on the W active site. According to the referee's suggestion, the water activation energy, binding energies of H^* intermediates on W-ACs without P doping were calculated for comparison. As shown in Supplementary Figs. 31, the energy barrier of the prior water dissociation on the sample without P doping (0.28 eV) is higher than that of P doped W-ACs (0.24

eV), and the positive value of H* intermediates suggested the weak binding energy of H* intermediates on the catalyst surface of W-ACs without P doping.

The number and position of P atom in W-SAs and W-ACs were determined by experimental characterization and previously reported DFT calculations: (i) according to the coupled plasma optical emission spectrometry (ICP-OES) analysis, the weight ratios of P and W elements were determined to be 2.57/9.67 and 2.62/10.49 wt% in W-SAs and W-ACs, respectively, with the molar ratios of W/P in W-SAs and W-ACs calculated to be 1:1.57 and 1:1.48, respectively (methods section), implying an extremely low mass loading of P, and only one or two P atoms were theoretically around the single tungsten atoms and tungsten clusters over the carbon substrates; (ii) the P 2p core level XPS spectrum of W-SAs and W-ACs indicated tungsten atoms in W-SAs and W-ACs were not directly coordinated with P atoms in the carbon matrix (Supplementary Fig. 18); (iii) according to the previous reports, the P-doping could accelerate the charge transfer during the alkaline HER process, due to the highly polarized P-C bond formed between P and C with different electronegativities, which will serve as an electron transfer bridge to improve the overall conductivity, and is beneficial for the water dissociation and the desorption of H*/OH* intermediates (*Nat. Commun.* **9**, 2533 (2018); *Adv. Sci.* **8**, 2101314 (2021)). However, the introduction of excessive P atoms into the established DFT models will decrease the stability because of the large structural distortion of the intrinsic carbon substrates, and thus previous DFT calculations involved in P doping usually illustrated the boosted electrocatalytic mechanism with similar models containing one P atom (*Nat. Commun.* **11**, 1853 (2020); *Sci. China Mater.* **63**, 965-971 (2020)). Meanwhile, the DFT models of one and two P atoms directly bonded to the W-W structures were also established in comparison with the optimized structure model in this work (Fig. 5b and Supplementary Fig. 29). As can be seen, large distortions of the carbon substrate were observed, moreover, the corresponding adsorption free energies (ΔG_{H}) of H* intermediate were calculated to be as high as -1.06 and -0.83 eV, respectively, suggesting they indeed afford inferior HER activities.

Supplementary Figure 31. The free energy diagrams for (a) water dissociation and (b) hydrogen desorption steps on W-ACs with and without P doping. (c) P 2p core level XPS spectra of W-SAs, W-ACs, and WC NPs. (d, e, f) Geometries of P-doped W-ACs and the corresponding adsorption free energies (ΔG_H) of possible active sites for H^* adsorption.

Comments 4:

In Fig 2c, the XPS valence band spectrum of W4f at 580 °C is highly closer to the Pt reference sample compared to that of W4f at 550 °C. But, the optimal temperature of ‘W-ACs’ synthesis is 550 °C in Methods. Why do authors set the synthesizing temperature at 550 °C for ‘W-ACs’? Is there any catalytic performance results using W-based single atom electrocatalyst treated in 580 °C ?

Author reply: Thanks for the referee’s valuable comments. Yes, the valence band profile at 580 °C is highly closer to the Pt reference compared to that of tungsten species collected at 550 °C, whereas we also observed that the W-C peaks of W 4f core-level XPS spectrum at 580 °C was slightly weaker than that of tungsten sample at 550 °C probably due to the aggregated W-ACs simultaneously coated by a graphitic carbon layer (*ACS Catal.* **2**, 765-769 (2012)), which would block the intrinsic W-C active sites, and affect the diffusion of electrolyte and the desorption of the produced hydrogen gas in the following alkaline HER electrocatalysis (*ACS Catal.* **2**, 765-769 (2012); *Appl. Catal. B: Environ.* **203**, 684-691 (2017)). Therefore, the optimized temperature range for synthesizing W-ACs was probably confined in 550-580 °C. The alkaline HER activities

of tungsten samples obtained at 550 and 580 °C were evaluated, respectively. As shown in Supplementary Fig. 19, the tungsten species obtained at 580 °C exhibited a markedly inferior alkaline HER activity in comparison with that of the sample at 550 °C with higher overpotential at 10 mA/cm² (91 mV) and larger Tafel slope (51 mV/dec), indicating the high-temperature induced graphitic carbon layer on tungsten species has an adverse effect on the electrocatalytic performance.

Supplementary Figure 19. (a) Polarization (LSV) curves and (b) Tafel plots of tungsten species synthesized at 550 and 580 °C, respectively.

Comments 5:

There are some typos or errors in the manuscript.

Line 21 in page 13, “relatively steady HER activity over 1 day”

Line 26 in page 18, “(Supplementary Fig. 9c)”

Figure 3 a has mis-labeled W-SAs with wrong color in legend (Blue -> Orange)

Author reply: Thanks for the referee’s valuable comments. We have carefully checked the manuscript, and the relevant typos and errors have been corrected in the revised manuscript.

REVIEWER COMMENTS

Reviewer #1 (Remarks to the Author):

The authors have made great efforts to respond to the queries of the reviewers and to further improve the data quality. I find these efforts satisfactory and can recommend publication of the manuscript.

Just three minor comments:

1. According to the figure caption, Figures 1(c) and 1(f) should contain red arrows. I cannot discern any such arrows. Moreover, I find it very difficult to identify the red circles in Figure 1(f).
2. The caption of Figure S13 mentions a green-shaded region. I cannot see any green-shaded region in the figure.
3. Supplementary Figure 10 is for an Ar/H₂ mixture at 0.1 mbar. What were the partial pressures of Ar and of H₂ (0.1 mbar Ar/0. 1 mbar H₂ and thus a total pressure of 0.1 mbar or a total pressure pressure of 0.1 mbar and lower partial pressure of Ar and H₂?)

Reviewer #2 (Remarks to the Author):

The authors have addressed some of my questions properly, but the explanation for Comment 3 is still inadequate. The observed W-O signals in experiments may be caused by the adsorption of OH⁻ in electrolyte rather than water dissociation, because the concentration of OH⁻ is very high on alkaline condition. In this case, the reaction $\text{H}_2\text{O}(\text{l}) + \text{e}^- \rightarrow \text{H}^* + \text{OH}^-(\text{l})$ seems to be more reasonable, because it involves an electron transfer and can be promoted by negative potential, while $\text{H}_2\text{O}^* \rightarrow \text{H}^* + \text{OH}^*$ is a thermochemical process independent on potential. Hence, I still strongly suggest the authors to calculate and discuss the barrier of $\text{H}_2\text{O}(\text{l}) + \text{e}^- \rightarrow \text{H}^* + \text{OH}^-(\text{l})$, compared with $\text{H}_2\text{O}^* \rightarrow \text{H}^* + \text{OH}^*$. The electrochemical barrier of alkaline Volmer reaction can be calculated with a "charge extrapolation method", using a water-layer model containing metal cation. You can reference ACS Catal. 2019, 9, 6194–6201.

Lamoureux, P. S.; Singh, A. R.; Chan, K. pH effects on hydrogen evolution and oxidation over pt(111): Insights from first-principles. ACS Catalysis 2019, 9, 6194-6201.

Reviewer #3 (Remarks to the Author):

The authors revised their manuscript substantially, which is now much improved and much more clear. The reviewer also acknowledged the authors for putting lots of effort to it. Now, I recommend this for the publication.

Answer to reviewer's comments:

We would like to thank the editor and reviewers for their deep and thorough reviewing of our manuscript. In view of these valuable queries, we have further revised our manuscript, with the changes marked out in the main text and also listed as follows. Here are the detailed responses to the comments.

Replies on comments of reviewer 1:

The authors have made great efforts to respond to the queries of the reviewers and to further improve the data quality. I find these effort satisfactory and can recommend publication of the manuscript.

Author reply: We appreciate the reviewer's recommendation very much.

Just three minor comments:

Comments 1:

According to the figure caption, Figures 1(c) and 1(f) should contain red arrows. I cannot discern any such arrows. Moreover, I find it very difficult to identify the red circles in Figure 1(f).

Author reply: Thanks for the referee's valuable comments. Much clearer symbols replacing the red arrows and red circles have been used to relabel the as-obtained tungsten species in Fig. 1c and f, and the corresponding figure captions have also been revised.

Fig. 1 (c) Enlarged STEM image of W-SAs. Inset shows the intensity profile extracted from the tungsten atoms indicated by the yellow solid circle. (f) Enlarged STEM image of W-ACs, with the W-W structures indicated by the yellow dashed circles.

Comments 2:

The caption of Figure S13 mentions a green-shaded region. I cannot see any green-shaded region in the figure.

Author reply: Thanks for the referee's valuable comments. We have relabeled the green-shaded region in Supplementary Fig. 13.

Supplementary Figure 13. Valence band of bare Ni foam and that coated with pristine P-doped carbon. The intensity of the green-shaded region near the Fermi level is very sensitive to the electronic structure of the sample, hence, compared to those of the bare Ni foam and C@Ni foam, the stronger signals of the W-SAs treated at 580 and 600 °C can be attributed to the aggregation of W atoms rather than the substrates. The valence band profiles of the reference samples (bare Ni foam and P-doped C@Ni foam) were obtained in the analysis chamber of NAP-XPS system (5×10^{-9} mbar), the valence band profiles of tungsten species at 550 and 580 °C were directly extracted from Fig. 2c.

Comments 3:

Supplementary Figure 10 is for an Ar/H₂ mixture at 0.1 mbar. What were the partial pressures of Ar and of H₂ (0.1 mbar Ar/0.1 mbar H₂ and thus a total pressure of 0.1 mbar or a total pressure pressure of 0.1 mbar and lower partial pressure of Ar and H₂?)

Author reply: Thanks for the referee's valuable comments. The total pressure of Ar/H₂ mixture is 0.1 mbar, and the partial pressure of Ar (90%) and H₂ (10%) was controlled by two independent leak valves. The relevant description has been added in the figure caption in Supplementary Fig. 10.

Replies on comments of reviewer 2:

The authors have addressed some of my questions properly, but the explanation for Comment 3 is still inadequate. The observed W-O signals in experiments may be caused by the adsorption of OH⁻ in electrolyte rather than water dissociation, because the concentration of OH⁻ is very high on alkaline condition. In this case, the reaction $\text{H}_2\text{O} (\text{l}) + \text{e}^- \rightarrow \text{H}^* + \text{OH}^- (\text{l})$ seems to be more reasonable, because it involves an electron transfer and can be promoted by negative potential, while $\text{H}_2\text{O}^* \rightarrow \text{H}^* + \text{OH}^*$ is a thermochemical process independent on potential. Hence, I still strongly suggest the authors to calculate and discuss the barrier of $\text{H}_2\text{O} (\text{l}) + \text{e}^- \rightarrow \text{H}^* + \text{OH}^- (\text{l})$, compared with $\text{H}_2\text{O}^* \rightarrow \text{H}^* + \text{OH}^*$. The electrochemical barrier of alkaline Volmer reaction can be calculated with a “charge extrapolation method”, using a water-layer model containing metal cation. You can reference ACS Catal. 2019, 9, 6194–6201.

Lamoureux, P. S.; Singh, A. R.; Chan, K. pH effects on hydrogen evolution and oxidation over Pt(111): Insights from first-principles. ACS Catalysis 2019, 9, 6194-6201.

Author reply: Thanks for the referee’s valuable comments. In order to address the referee’s concerns comprehensively and accurately, we have divided above comments into three queries, in which a point-by-point response has been made as follows.

Comments 1:

The observed W-O signals in experiments may be caused by the adsorption of OH⁻ in electrolyte rather than water dissociation, because the concentration of OH⁻ is very high on alkaline condition.

Author reply: Thanks for the referee’s valuable comments. In fact, in order to eliminate the adsorption interference of OH⁻ anions in the electrolyte, W 4f core-level XPS spectra of W-ACs only soaked in KOH solution without electrochemical treatment (initial) has been measured as the baseline for evaluating the change in W-O signals of the W-AC samples treated by the increased negative potentials. As shown in Fig. 4b, W-ACs exhibited inevitable W-O (W-O/W-C(O)) signals at high binding energies after deconvolution, which can be attributed to the adsorbed OH⁻ (electrolyte) and robust oxygen species in the initial state. Interestingly, the W-ACs treated at 0.00 V versus RHE have an obviously enhanced concentration of W-O (W-O/W-C(O)) bonds compared with the sample treated only by soaking in KOH solution (Supplementary Table 6), indicating the applied potential was too low to repel the adsorbed OH^{*} intermediates in the alkaline HER process. It should be noted that the enhanced W-O (W-O/W-C(O)) signals are caused by water

dissociation rather than the adsorption of OH⁻ anions in electrolyte at 0.00 V, which can be understood by the following two reasons: (i) the W-AC-catalyzed alkaline HER process takes place at the cathode, which will hinder the adsorption of the OH⁻ anions (electrolyte) because of the Coulomb effect during the process of alkaline HER; (ii) a large amount of electrons are accumulated at the cathode during the process of alkaline HER, which will protect the tungsten atoms from oxidation to higher valence state, implying the W-O signals detected on used W-ACs were not stronger than that of the initial state when the applied negative potentials were removed. As expected, the intensities of the signals corresponding to W-O (W-O/W-C(O)) bonds markedly reduced with enlarged negative potentials from -0.02 to -0.06 V (Supplementary Table 6), indicating the adsorbed OH* intermediates could be easily detached from the W sites into the electrolyte at low potentials, and no more OH⁻ anions (electrolyte) were adsorbed on the catalyst surface in comparison with the initial state when the applied negative potentials were removed. Accordingly, our experimental observations have provided direct evidence for the adsorption/desorption of OH* intermediates. The relevant discussion has been added in Fig. 4 in the revised manuscript.

Fig. 4 (b) W 4f core level XPS spectra of W-ACs treated at 0.00, -0.02, -0.04, and -0.06 V versus RHE and W-ACs soaked in KOH solution, respectively. (c) The corresponding values of the peak intensity extracted from the W 4f core level XPS spectra in Fig. 4b.

Comments 2:

In this case, the reaction $\text{H}_2\text{O} (\text{l}) + \text{e}^- \rightarrow \text{H}^* + \text{OH}^- (\text{l})$ seems to be more reasonable, because it involves an electron transfer and can be promoted by negative potential, while $\text{H}_2\text{O}^* \rightarrow \text{H}^* + \text{OH}^*$ is a thermochemical process independent on potential.

Author reply: Thanks for the referee's valuable comments. Yes, we agree well with the referee's viewpoint that the reaction $\text{H}_2\text{O} (\text{l}) + \text{e}^- \rightarrow \text{H}^* + \text{OH}^- (\text{l})$ involves an electron transfer and can be promoted by negative potential, while $\text{H}_2\text{O}^* \rightarrow \text{H}^* + \text{OH}^*$ is a thermochemical process independent on potential. However, Duan, et al. reported a pioneering work that demonstrated a superior alkaline HER activity using $\text{Ni}(\text{HCO}_3)_2/\text{MoS}_2$ heterostructure catalyst (*Chem. Commun.*, **56**, 12065-12068 (2020)), in which OH^* intermediates from initial water dissociation (thermochemical process) were adsorbed on the active material of $\text{Ni}(\text{HCO}_3)_2$, followed by release of the OH^- anions into the electrolyte (electrochemical water dissociation). This process was evidenced to be the optimized reaction pathway in alkaline Volmer step by DFT calculations. Clearly, such a step-wise evolution process demonstrated that the produced hydroxyl species have underwent the adsorption (OH^*) and subsequent detaching (OH^-) processes in alkaline Volmer reaction, which were simultaneously controlled by the thermochemical and electrochemical kinetics. In addition, we would like to point out a small detail that there is a difference for the adsorption model of H_2O molecules in alkaline HER process when using atomic-dispersion catalysts and bulk metal materials, respectively. For atomic-dispersion alkaline HER catalysts, H_2O reactants are usually adsorbed on the catalytic sites (*) in the initial step of such processes (denoted as H_2O^* or H_2O^{+*}), as confirmed by numerous HER experiments and DFT calculations (*Nat. Catal.* **2**, 134-141 (2019); *Nat. Commun.* **12**, 3021 (2021); *Nano Lett.* **20**, 8375-8383 (2020)). However, a interfacial water-layer model (denoted as H_2O (liquid)) containing metal (Na^+ , K^+) cation with distance of $\sim 4.0 \text{ \AA}$ to the catalyst surface is universal for bulk noble metal catalyzed alkaline HER process (*Nature* **600**, 81-85 (2021); *ACS Catalysis* **9**, 6194-6201 (2019)). Such a small difference is probably attributed to the unsaturated coordination environment of atomic-dispersed sites, because atomic-dispersion metal centers with low-coordination structure usually have stronger ability to adsorb much more reactants in comparison with that of the bulk counterparts in the initial step of catalysis (*Acc. Chem. Res.* **46**, 1740-1748 (2013); *Chem. Soc.*

Rev., **48**, 5310-5349 (2019)). In particular, tungsten as a 5d transition metal in group six of the periodic table of elements is remarkable for its complex electronic structure featuring open d and f shells (*Nat. Chem.* **3**, 336 (2011)), in which the abundant outermost vacant orbitals will result in more H₂O reactants being adsorbed on the exposed W-W dual-sites (*Nat. Commun.* **10**, 2149 (2019)).

Comments 3:

Hence, I still strongly suggest the authors to calculate and discuss the barrier of H₂O (l) + e⁻ → H* + OH⁻ (l), compared with H₂O* → H* + OH*. The electrochemical barrier of alkaline Volmer reaction can be calculated with a “charge extrapolation method”, using a water-layer model containing metal cation. You can reference *ACS Catal.* 2019, 9, 6194–6201.

Lamoureux, P. S.; Singh, A. R.; Chan, K. pH effects on hydrogen evolution and oxidation over Pt(111): Insights from first-principles. *ACS Catalysis* 2019, 9, 6194-6201.

Author reply: Thanks for the referee’s valuable comments. In order to address the referee’s concern on the barriers of above-mentioned two reaction pathways, much effort has been devoted to calculate and discuss the electrochemical barriers of H₂O (l) + e⁻ → H* + OH⁻ (l) and H₂O* → H* + OH* in accordance with the water-layer model containing metal cation (*ACS Catalysis* **9**, 6194-6201 (2019); *J. Phys. Chem. Lett.* **6**, 2663-2668 (2015)). As expected, no matter what reaction pathway was followed, no energy barriers were calculated for the cleavage of H-OH bonds on W-W dual sites (Supplementary Fig. 32), indicating W centers indeed have strong ability to break H-OH bonds in alkaline Volmer step. Then, the produced H* intermediates were adsorbed on the W sites, whereas the split OH⁻ radicals were directly retained in the electrolyte without desorption process based on the reaction pathway suggested by the referee. However, the adsorbed OH* intermediates will be repelled into the electrolyte when increasing applied potentials based on our experimental observations (Fig. 4). Note that the realistic electrocatalytic environment is much more complicated than that of the established DFT model, and thereby much more powerful analytic techniques are required for a further comprehension of the catalytic mechanism (*Nat. Catal.* **1**, 165-166 (2018)), but our DFT prediction and aforementioned experimental observations have evidenced that the reaction pathway of H₂O* → H* + OH* and subsequent desorption of OH* should also be reasonable. The relevant description has been added in the revised manuscript as follows:

“which was further evidenced by the calculations based on a water-layer model containing metal cation^{52,53} (Supplementary Fig. 32).”

Supplementary Figure 32. The cleavage of H-OH bonds on W-W dual sites in accordance with reaction pathways of (a) $\text{H}_2\text{O}(\text{l}) + \text{e}^- \rightarrow \text{H}^* + \text{OH}^-$ and (b) $\text{H}_2\text{O}^* \rightarrow \text{H}^* + \text{OH}^*$, where water-layer model containing metal cation was established for calculating the electrochemical barriers of above-mentioned reaction pathways.

Replies on comments of reviewer 3:

The authors revised their manuscript substantially, which is now much improved and much more clear. The reviewer also acknowledged the authors for putting lots of effort to it. Now, I recommend this for the publication.

Author reply: We appreciate the reviewer’s recommendation very much.

REVIEWERS' COMMENTS

Reviewer #2 (Remarks to the Author):

The authors have addressed my concerns and I recommend the publication of this manuscript.

Replies on comments of reviewer 2:

Comments 1:

The authors have addressed my concerns and I recommend the publication of this manuscript.

Author reply: We appreciate the reviewer's recommendation very much.